# RagB stimulates the activity of the peptidoglycan polymerase RodA in *Bacillus subtilis*

Frédérique Pompeo [1✉], Elodie Foulquier[1], Arnaud Chastanet [2], Leon Espinosa[1], Cyrille Billaudeau [2], Anthony Rodrigues [1], Charlène Cornilleau [2], Rut Carballido-López [2] & Anne Galinier [1]

## Abstract

**The bacterial cell wall is primarily composed of peptidoglycan (PG), a polymer essential for its protective envelope function, and any defect in its synthesis or repair can potentially result in bacterial lysis. Class A Penicillin-Binding Proteins (aPBPs) and Shape, Elongation, Division, and Sporulation (SEDS) proteins are PG polymerases acting in concert to ensure bacterial cell wall growth. Here, we identify the first regulator of the SEDS protein RodA in the Gram-positive model bacterium *Bacillus subtilis*. In the presence of the antibiotic moenomycin, which specifically inhibits glycosyl-transferase activity of aPBPs, or in a strain deleted for all four aPBPs, bacterial survival depends on the presence of the YrrS protein (renamed RagB) and can be rescued by overexpression of RodA. No effect of RagB is observed on the *rodA* gene expression level or on the speed of circumferentially moving RodA associated with PG elongation by the Rod complex. However, we demonstrate that RagB interacts with RodA. We propose that RagB stimulates RodA activity and becomes essential in the absence of aPBPs and in particular of the major aPBP, PBP1.**

**Keywords** Peptidoglycan Synthesis; aPBP; SEDS; *Bacillus subtilis*; Regulation
**Subject Categories** Microbiology, Virology & Host Pathogen Interaction; Signal Transduction

## Introduction

Peptidoglycan (PG) is the main component of the bacterial cell wall (CW). It is a polymeric three-dimensional meshwork that ensures bacterial cell shape and resistance to osmotic pressure and various environmental stresses (Egan et al, 2020). Because PG is essential and unique to bacterial cells, it is a prime target for many current antibiotics, most of which act on the enzymes of its biosynthesis (Sauvage and Terrak, 2016). Regulators controlling the activity of these enzymes are therefore potential new targets in the race to find new antibiotics for multi-resistant bacteria. PG consists of glycan strands alternating N-acetylglucosamine (GlcNAc) and N-acetylmuramic acid (MurNAc) residues linked by β-1,4 glycosidic bonds, and crosslinked together by peptide bridges (Barreteau et al, 2008). PG precursors are first synthesized in the cytoplasm. The final lipid-anchored product GlcNAc-MurNAc-pentapeptide (Lipid-II), which carries the PG monomer subunit, is then flipped across the plasma membrane and inserted into the existing PG meshwork by High Molecular Mass Penicillin-Binding Proteins (HMW PBPs) (Macheboeuf et al, 2006) and Shape, Elongation, Division and Sporulation (SEDS) proteins (Emami et al, 2017; Meeske et al, 2016; Zhao et al, 2017).

HMW PBPs are classified into two families: class A (aPBPs) possessing both glycosyltransferase (GT) and transpeptidase (TP) activities necessary for the synthesis of PG, and class B (bPBPs) possessing only TP activity. GT activity is also provided by SEDS proteins, which work in tandem with bPBPs (Meeske et al, 2016; Sjodt et al, 2020; Straume et al, 2021). The model Gram-positive rod-shaped bacterium *Bacillus subtilis* contains three proteins of the SEDS family involved in PG polymerization. RodA is the main GT involved in the synthesis of the cylindrical cell wall by the Rod complex, also named elongasome. The Rod complex, which includes RodA and its cognate bPBPs, PBP2A and PbpH, the bacterial actins MreB and Mbl and the morphogenetic MreC, MreD and RodZ proteins (Carballido-López and Formstone, 2007), moves circumferentially and bidirectionally around the cell periphery, oriented by MreB proteins and powered by PG synthesis itself (Domínguez-Escobar et al, 2011; Garner et al, 2011). It was recently shown that the dynamics of the elongasome motility is bidirectional thanks to pauses and reversals dependent on the cellular level of RodA (Middlemiss et al, 2024). The other two SEDS proteins of *B. subtilis* are FtsW, which provides GT activity to the divisome, the complex that synthesizes the septal PG crosswall during cell division (Bisson-Filho et al, 2017; Taguchi et al, 2019; Whitley et al, 2024; Yang et al, 2021), and SpoVE, which is required for spore cortex formation during sporulation (Real et al, 2008).

It has been proposed that aPBPs play a role in PG maintenance and repair (Brunet et al, 2022; Vigouroux et al, 2020), whereas SEDS-bPBP complexes are required for de novo PG synthesis during cell division, elongation and sporulation: FtsW-PBP2B, RodA-PBP2A/PbpH and SpoVE-SpoVD, respectively in *B. subtilis* (Cho et al, 2016; Emami et al, 2017; Fay et al, 2010; Taguchi et al, 2019). It has been shown that the combined action of the two

[1]Aix-Marseille Univ, CNRS, LCB, UMR 7283, IMM, Marseille, France. [2]Université Paris-Saclay, INRAE, AgroParisTech, Micalis Institute, Jouy-en-Josas, France.
✉E-mail: fpompeo@imm.cnrs.fr

systems (Rod complex and aPBPs) defines bacterial rod-morphology and their balanced action has been proposed to regulate cell width in *B. subtilis*; the Rod complex reduces diameter, whereas aPBPs increase it (Dion et al, 2019). The activities of both types of PG synthases must be finely regulated to keep the integrity of the PG throughout the cell cycle in order to ensure bacterial shape and survival. Some PBP regulators have been identified to date in both Gram-negative (Kermani et al, 2022; Paradis-Bleau et al, 2010; Sardis et al, 2021) and Gram-positive species (Delisle et al, 2021; Fenton et al, 2018; Lenoir et al, 2023; Morlot et al, 2013; Stamsås et al, 2020). In all cases, they act directly on PBPs by modifying either their activity or their cellular localization. More recently, it has been shown in both *Helicobacter pylori* (Contreras-Martel et al, 2017) and *Escherichia coli* (Liu et al, 2020; Rohs et al, 2018; Shlosman et al, 2023) that the essential MreC protein may orchestrate the dynamics of structural changes underlying the activity of the pair formed by RodA and PBP2A. This complex can adopt two conformational states, and it was proposed that MreC would favor the open active conformation with a return to an off conformation mediated by an interaction between PBP2A and MreD (Liu et al, 2020). MreC and MreD components of the Rod complex are conserved and may thus serve a similar function in *B. subtilis*. Furthermore, the strong redundancy of PBPs (four aPBPs and six bPBPs) in *B. subtilis* suggests that a precise regulation of each of their activity probably exists during changes of growth conditions, and that the inactivation of one could be compensated for by the activation of another (Mitchell et al, 2024).

Seeking new regulators of PBPs or SEDS, we considered that they could be part of the protein complexes involved in PG polymerization during cell elongation or division. One interesting protein is GpsB, which acts as a platform for proteins involved in these complexes (divisome and elongasome) (Cleverley et al, 2019) and regulates septal and lateral PG synthesis in low-GC Gram-positive bacteria (Costa et al, 2024; Fleurie et al, 2014). In *B. subtilis*, GpsB has been shown to facilitate the removal of PBP1 from the cell pole after its complete maturation (Claessen et al, 2008), and proposed direct interactors of GpsB include MreC, RodZ, several PBPs and two proteins of unknown function, YpbE and YrrS (Cleverley et al, 2019). Both YpbE and YrrS were found to have an interaction motif with the cytoplasmic domain of GpsB, similar to the one identified in PBP1 (Cleverley et al, 2019). They are transmembrane proteins with an extracellular region composed of an intrinsically disordered region (IDR) and a C-terminal folded domain. For YpbE, this domain resembles the LysM domains, which have been described as capable of binding PG (Pereira et al, 2019), but the YrrS C-terminal domain has not been characterized. Recently, a genetic screening of genes involved in cell division in *B. subtilis* by double CRISPRi identified *yrrS* and *ypbE* as two candidates with negative genetic interactions with *ezrA* but not with *gpsB* (Koo et al, 2024). EzrA is a negative regulator of Z-ring formation, and its absence leads to a multiple Z-rings phenotype at mid-cell (Adams and Errington, 2009). It also recruits PBP1 at the division septum (Claessen et al, 2008). *yrrS* or *ypbE* deletions alone did not exhibit morphological defects, but upon *ezrA* deletion, they increased the filamentation phenotype of the Δ*ezrA* strain (Koo et al, 2024).

To investigate whether YpbE and YrrS could be potential regulators of PG synthases, deletion mutants were constructed and their growth and morphology analyzed. Whereas deletion of *ypbE* has a mild effect, we demonstrated that in the absence of the main aPBP, PBP1, the YrrS protein, which we have renamed RagB (for <u>R</u>od<u>A</u> <u>G</u>T <u>B</u>ooster), is required to maintain cell growth and morphology. We also found that overproduction of the SEDS protein RodA suppresses the essentiality of RagB in this genetic background. We showed as well that RagB is not involved in the regulation of *rodA* expression or on the speed of the circumferentially moving RodA molecules associated with the Rod complex. Instead, our results suggest that RagB stimulates RodA activity via a protein-protein interaction and that cell morphology and survival of a strain that no longer produces aPBPs depend strictly on the presence of RagB.

# Results

## RagB is necessary for growth and morphology in the absence of PBP1

In *B. subtilis*, GpsB was first described as a protein involved in the control of the cell cycle by shuttling between the septum and the lateral CW together with PBP1 (Claessen et al, 2008). A later study proposed that GpsB serves as an adapter for proteins of the elongasome and the divisome (Cleverley et al, 2019); GpsB interacting with several CW enzymes and membrane proteins. In order to test whether YpbE and YrrS (RagB), two proteins of unknown function previously found associated with GpsB (Cleverley et al, 2019), could have a regulatory role for PG enzymes with which they are linked, we first constructed some strains deleted for *gpsB*, *ponA* (the gene encoding PBP1) or both genes and monitored their growth rate and morphology in LB-rich medium (Fig. 1). As previously described, the Δ*gpsB* strain displayed wild-type growth and morphology (Fig. 1A,B) and the Δ*ponA* strain a slight growth defect associated with a thinner cell morphology (Fig. 1A,C) (Patel et al, 2020; Pedersen et al, 1999; Popham and Setlow, 1996). However, the Δ*ponA* Δ*gpsB* mutant displayed a strong growth defect, and cells were thinner than Δ*ponA* cells (Fig. 1A,C), sometimes with a slightly curved appearance (Fig. 1B). Altogether, these observations suggest that the role of GpsB extends beyond its interaction with PBP1; in other words, GpsB and aPBPs play significant additive roles in CW synthesis.

We then did the same analysis for strains containing the deletion of *ragB* (formerly *yrrS*) or *ypbE*, alone and combined with the Δ*ponA* mutation (Fig. 1; Appendix Fig. S1). The Δ*ragB* and Δ*ypbE* single mutants displayed wild-type growth and morphology, while deletion of *ypbE* in a Δ*ponA* background slightly exacerbated the Δ*ponA* mutant phenotypes (Appendix Fig. S1). However, the Δ*ragB* Δ*ponA* mutant had strong growth and morphological defects (Fig. 1A,B). We thus decided to characterize the role of RagB in this study. The Δ*ragB* Δ*ponA* strain grew very slowly and to a lower final optical density (Fig. 1A), and cells were irregular, curved and thinner than wild-type cells (Fig. 1B,C). These phenotypes are similar to those of the Δ*ponA* Δ*gpsB* mutant, suggesting a potential role for RagB in CW synthesis and possibly that GpsB and RagB could act on the same pathway. We then analyzed the relationship between RagB and GpsB, and tested whether overproduction of GpsB or RagB could complement the growth of the Δ*ponA* Δ*ragB* and the Δ*ponA* Δ*gpsB* mutants, respectively. We thus constructed strains that overproduce GpsB or RagB by xylose induction, in each

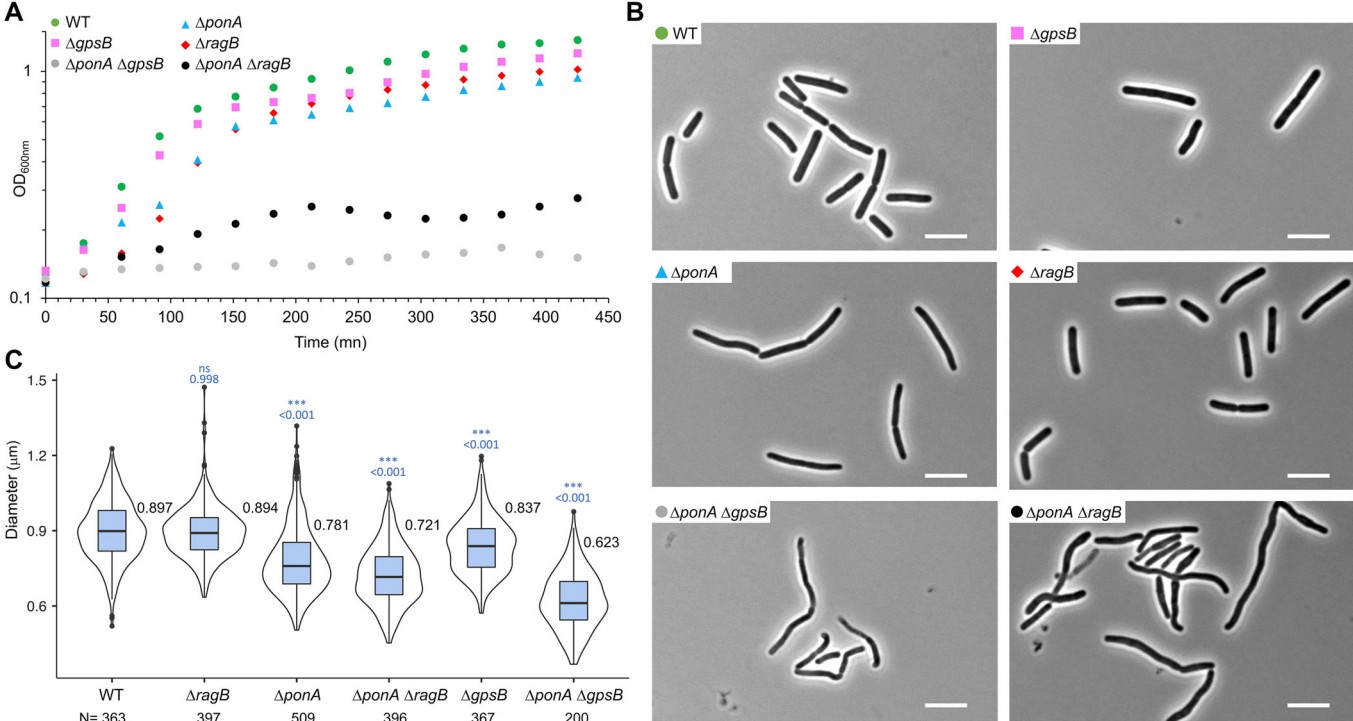

**Figure 1. Growth and morphology of strains deleted for *ponA*, *gpsB* and or *ragB* genes.**

(A) Growth of the different strains was tested on a 96-well plate in triplicate and incubated on a microplate reader at 37 °C for 7 h with stirring and OD measurements every 30 min. The representative growth curves of three biological replicates are shown. Growth of strains wild-type 168 (WT) in green, PS2062 (*ponA::spec*) in blue, JR46 (*gpsB::kan*) in pink, SG314 (*ponA::spec gpsB::kan*) in gray, BKE27300 (*ragB::erm*) in red and SG667 (*ponA::spec ragB::erm*) in black. (B) Microscopy images of strains wild-type 168 (WT) in green, PS2062 (*ponA::spec*) in blue, BKE27300 (*ragB::erm*) in red, SG667 (*ponA::spec ragB::erm*) in black, JR46 (*gpsB::kan*) in pink and SG314 (*ponA::spec gpsB::kan*) in gray cultured in LB at 37 °C up to OD₆₀₀ₙₘ = 0.3 were taken to observe their cell morphology. Scale bar is 5 μm. (C) Cell diameters of mutant strains were measured from microscopy images of strains wild-type 168, BKE27300 (*ragB::erm*), PS2062 (*ponA::spec*), SG667 (*ponA::spec ragB::erm*), JR46 (*gpsB::kan*) and SG334 (*ponA::spec gpsB::kan*) cultured in LB at 37 °C up to OD₆₀₀ₙₘ = 0.4. Violin plots of cell diameter (μm) whose average value is shown at top right, with *N* indicates the number of cells analyzed for each strain from three biological replicates. Blue boxes are box plots and dark lines indicate the median. Diameters are measured with the MicrobeJ plug-in. Statistically significant differences between the wild-type and mutant diameters are determined with ANOVA using a Tukey's test and exact *P* values are indicated. Data information: In (C), data are presented as Violin and box plots. Each box represents the interquartile range (IQR), with the lower and upper bounds corresponding to the 25th and 75th percentiles, respectively. The center line indicates the median (50th percentile). Whiskers extend to the minimum and maximum values within 1.5 × IQR; data points beyond this range are considered outliers. ns not significant, ***$P < 0.001$ (Tukey's test). Source data are available online for this figure.

mutant background and monitored their growth in parallel (Appendix Fig. S2A,B). The xylose-inducible GpsB and RagB proteins were produced and functional since they restored the growth defects of the Δ*ponA* Δ*gpsB* and Δ*ponA* Δ*ragB* strains, respectively (Appendix Fig. S2A,B). However, neither the overproduction of GpsB nor RagB was able to compensate for the absence of the other, suggesting either different cellular functions or an interdependence of these proteins (i.e., one cannot fully function in the absence of the other).

## The absence of RagB is compensated by the overproduction of RodA

As mentioned in the introduction, PBP1 and RodA work in concert to synthesize PG along the sidewall and in the absence of one, the other ensures all PG synthesis (Meeske et al, 2016). In agreement with this, *rodA* is overexpressed in a Δ*aPBP* mutant (Emami et al, 2017) and artificial overproduction of RodA compensates for the loss of aPBPs in cell growth (Kawai et al, 2023; Meeske et al, 2016). The lack of RagB had no apparent effect in a wild-type background

but had strong growth and morphological effects in the absence of PBP1, suggesting that RagB could function in the same pathway as RodA. Thus, we decided to test if an artificial overproduction of RodA could complement the phenotypes of the Δ*ragB* Δ*ponA* mutant. We constructed a strain overproducing RodA by xylose induction from the ectopic *amyE* locus and monitored its growth alongside that of the strain overproducing RagB used as a positive control. As shown in Fig. 2A, the addition of xylose had no effect on the wild-type, Δ*ponA* or Δ*ponA* Δ*ragB* strains, but it rescued the growth defects of the Δ*ponA* Δ*ragB* strains overexpressing either *ragB* or *rodA* under P_xyl promoter. In addition, the overproduction of RodA (or RagB) drastically reduced the shape defects observed in the absence of *ponA* and *ragB* (Fig. 2B). These results indicate that an excess of RodA can reduce the phenotypes of Δ*ragB* Δ*ponA* mutant to those of Δ*ponA* mutant (Fig. 2A,B). Overproduction of a RodA variant carrying a mutation in its catalytic site that greatly reduces its GT activity (Meeske et al, 2016), RodA(D280A), did not complement the growth and morphology phenotypes of the Δ*ponA* Δ*ragB* mutant (Fig. 2A,B), indicating that the presence of an active RodA protein is essential to compensate for the effects of RagB

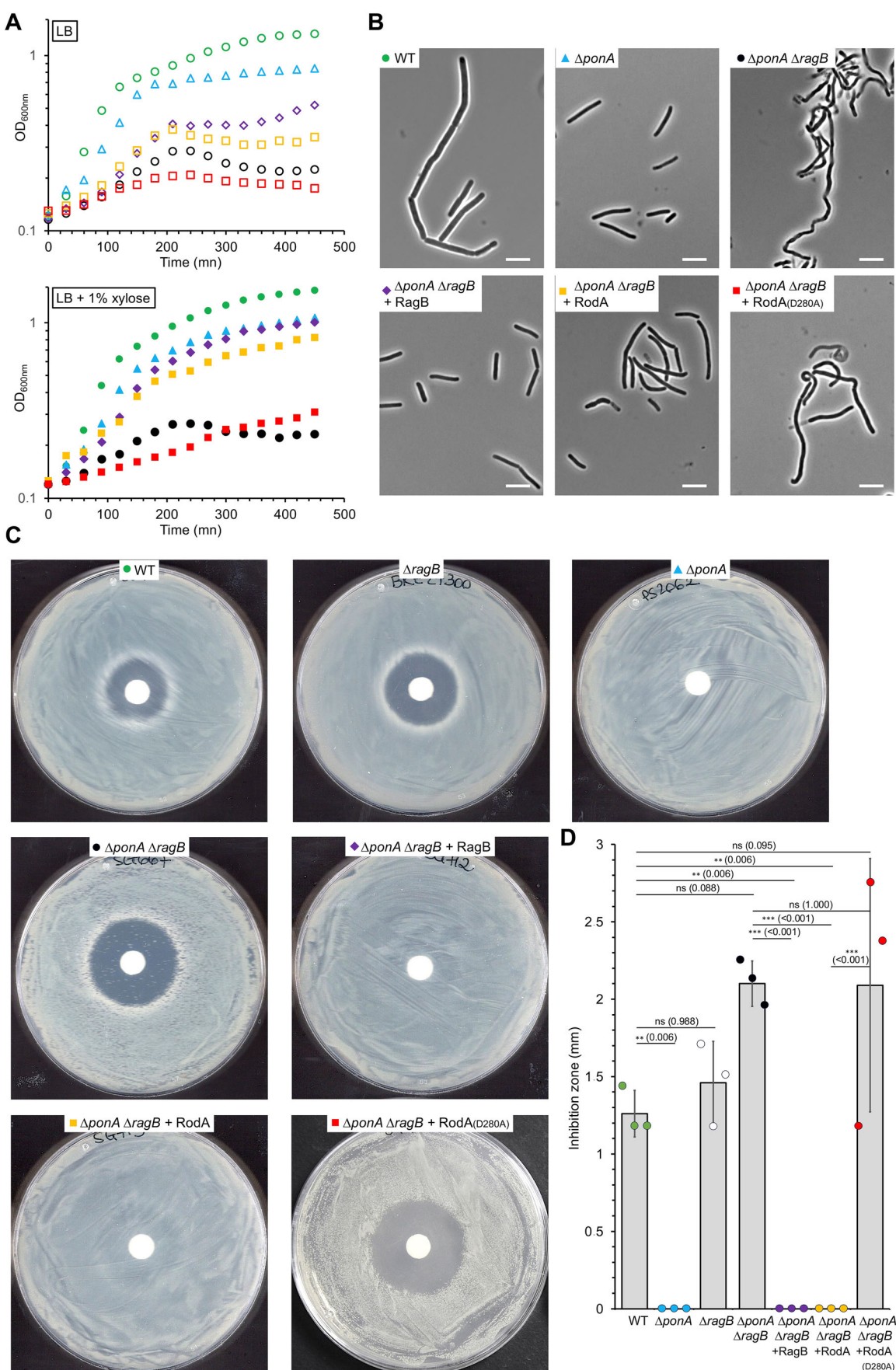

**Figure 2. Complementation of SG667 (*ponA::spec ragB::erm*) phenotypes by RodA overproduction.**

(A) Growth of the different strains was tested in triplicate on a 96-well plate at 37 °C for 7 h. Growth in LB (top graph, empty symbols) and LB + 1% xylose (bottom graph, solid symbols) of the strains wild-type 168 (WT) in green, PS2062 (*ponA::spec*) in blue, SG667 (*ponA::spec ragB::erm*) in black, SG712 (*ponA::spec ragB::erm amyE::*$P_{xyl}$*ragB*) in purple, SG713 (*ponA::spec ragB::erm amyE::*$P_{xyl}$*rodA*) in yellow and SG933 (*ponA::spec ragB::erm amyE::*$P_{xyl}$*rodA(D280A)*) in red are presented on the graph. The representative growth curves of three biological replicates are shown. (B) Microscopy images of strains WT in green, PS2062 (*ponA::spec*) in blue, SG667 (*ponA::spec ragB::erm*) in black, SG712 (*ponA::spec ragB::erm amyE::*$P_{xyl}$*ragB*) in purple, SG713 (*ponA::spec ragB::erm amyE::*$P_{xyl}$*rodA*) in yellow and SG933 (*ponA::spec ragB::erm amyE::*$P_{xyl}$*rodA(D280A)*) in red, cultured in LB at 37 °C up to $OD_{600nm} = 0.3$, were taken to analyze their cell morphology. Scale bar is 5 μm. (C) Strains were grown at 37 °C on LB medium supplemented with 15 mM $MgSO_4$ to an $OD_{600nm}$ of 0.4. A 800 μl aliquot of each culture was centrifuged for 3 min at 7000 rpm. Cell pellets were resuspended in 300 μl of LB, and 200 μl were spread on a LB-Agar plate containing 1% xylose. In total, 40 μg of moenomycin were deposited on filter paper disks placed in the center of the plates that were incubated at 30 °C overnight. Pictures were taken to compare the size of the inhibition zone for each strain: wild-type 168 (WT), BKE27300 (*ragB::erm*), PS2062 (*ponA::spec*), SG667 (*ponA::spec ragB::erm*), SG712 (*ponA::spec ragB::erm amyE::*$P_{xyl}$*ragB*), SG713 (*ponA::spec ragB::erm amyE::*$P_{xyl}$*rodA*) and SG933 (*ponA::spec ragB::erm amyE::*$P_{xyl}$*rodA(D280A)*). All the experiments were realized in triplicate and representative experiments are shown here. (D) Bar graph of the zone of growth inhibition (in mm) by moenomycin for the strains shown in (C). Each obtained value from the three biological replicates is represented by a plot. Error bars of the graph are the SD. Statistically significant differences between the wild-type and mutant inhibition diameters were determined with ANOVA using a Tukey's test and exact *P* values are indicated. Data information: In (D), data are presented as mean ± SD. ns not significant, **$P < 0.01$, ***$P < 0.001$ (Tukey's test). Source data are available online for this figure.

absence on Δ*ponA* cell growth and morphology. Furthermore, the absence of GpsB in a Δ*ponA* context was not compensated for by the overproduction of RodA (Appendix Fig. S2C). This result does not allow us to discriminate between the two possibilities: either RagB and GpsB have distinct roles or they may function together. However, in this case, since GpsB has a pleiotropic role and serves as an adapter for multiple CW enzymes, including RagB (Cleverley et al, 2019), the overproduction of RodA is not sufficient to compensate for the absence of GpsB in a Δ*ponA* context. In addition, we compared the growth of the Δ*ponA* Δ*ragB* Δ*gpsB* mutant with that of the Δ*ponA* Δ*gpsB* and Δ*ponA* Δ*ragB* strains (Appendix Fig. S3) and observed an additive effect of the deletions. This suggests that the two proteins RagB and GpsB, although interacting together, have actually specific roles and may be involved in different pathways. However, that doesn't mean they cannot also be involved in a common function. Complementation of the Δ*ponA* Δ*ragB* Δ*gpsB* mutant by overproduction of PBP1, RagB, GpsB or RodA showed that only the overproduction of PBP1 allowed a return to normal growth, as expected (Appendix Fig. S3).

To determine whether RodA overproduction compensates for the absence of RagB or only for the decrease in GT activity associated to the absence of PBP1, we performed moenomycin resistance tests (Fig. 2C,D). Moenomycin inhibits the GT activity of aPBPs but not of SEDS (Emami et al, 2017). We observed that the Δ*ragB* mutant was slightly more sensitive to moenomycin than the wild-type strain; in such mutant, RodA activity is basal. In a Δ*ponA* strain lacking PBP1, the main GT activity is carried by RodA that is upregulated and possibly boosted by RagB, which makes bacteria resistant to the antibiotic (Emami et al, 2017) (Fig. 2C,D). The Δ*ragB* Δ*ponA* mutant was more sensitive to moenomycin than the Δ*ragB* mutant and the wild-type strain (Fig. 2C,D). In the absence of *ragB* and *ponA*, RodA activity is insufficient to support growth when the remaining aPBPs are inactivated by the moenomycin. This could suggest that a larger fraction of the PG synthesis activity is supported by the other aPBPs driving to an increased sensitivity to the antibiotic. Interestingly, this Δ*ragB* Δ*ponA* strain became resistant when RagB (positive control) or active RodA was overproduced (Fig. 2C,D). These strains had thus the same phenotype as a Δ*ponA* mutant, showing that the overproduction of RodA compensates for the absence of RagB. Once again, when the RodA(D280A) mutant was overproduced, complementation was no longer observed, indicating that RodA GT activity was

necessary (Fig. 2C,D). Taken together, these results suggest that the defect in growth and morphology of the Δ*ragB* Δ*ponA* mutant may be explained by a reduction in RodA activity due to the absence of RagB.

## RagB does not affect the expression level of *rodA*

A potential reduction of RodA GT activity in Δ*ragB* cells could be explained by several ways. The simplest hypotheses are that, in the absence of RagB, either RodA levels are lowered, or RodA activity is reduced, resulting from abnormal localization or cellular dynamics of RodA, or from a direct decrease of its enzymatic activity. To first test if the absence of *ragB* affects *rodA* expression, we carried out quantitative RT-PCR experiments in the following four strains: wild-type, Δ*ponA*, Δ*ragB* and Δ*ragB* Δ*ponA* (Fig. 3A). As expected, *rodA* expression level was increased in the Δ*ponA* background due to σ$^M$ upregulation (Emami et al, 2017; Patel et al, 2020). However, *rodA* expression level was unchanged in the Δ*ragB* mutant compared to the wild-type strain. The level of *ragB* expression was itself slightly increased in the Δ*ponA* mutant (Fig. 3B). We then compared RagB protein levels between the Δ*ponA* mutant and a wild-type strain by western blot and found that the variation of *ragB* expression observed had little effect on the cellular level of RagB protein (Fig. 3C). We concluded that *rodA* expression is not affected by RagB, and that *rodA* overexpression in the Δ*ponA* background is not linked to a variation in the amount of RagB protein. The absence of RagB does thus not modify the expression of *rodA*.

In order to check if the absence of RagB affects RodA protein level, a western blot was made in wild-type, Δ*ponA*, Δ*ragB* and Δ*ponA* Δ*ragB* strains producing Halo-tagged RodA (Appendix Fig. S4). Despite poor quality detection, these data seem to confirm that RodA was more produced in a Δ*ponA* but not in a Δ*ragB* context; further demonstrating that *rodA* is not subject to a posttranscriptional control by RagB.

## RagB interacts with RodA in vitro and in vivo

A second hypothesis that could explain the functional link between RodA and RagB is the modulation of RodA GT activity by RagB. This could occur via a direct interaction or via protein intermediate(s). To explore this question, we first performed

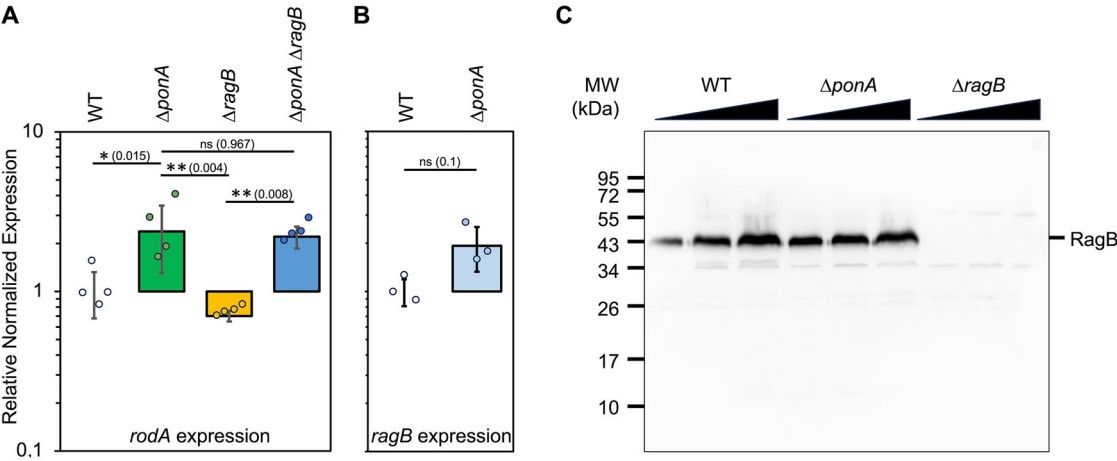

**Figure 3. *rodA* expression is not affected by RagB.**

(A) Bar graph of *rodA* expression levels in the strains wild-type 168 (WT), PS2062 (*ponA::spec*), BKE27300 (*ragB::erm*) and SG667 (*ponA::spec ragB::erm*) grown in LB medium until $OD_{600nm} = 0.5$ at 37 °C. The expression level was determined by quantitative RT-PCR and represented (as colored bars) for the three strains relative to that of the WT strain (value = 1). Four biological replicates were realized and each obtained value is indicated by a plot; error bars represent the SD. The statistically significant differences between wild-type and mutant strains were determined with ANOVA using a Tukey's test and the *P* values are indicated. (B) Bar graph of *ragB* expression levels quantified by quantitative RT-PCR for the strains wild-type 168 (WT) and PS2062 (*ponA::spec*) grown in LB medium until $OD_{600nm} = 0.5$ at 37 °C. Each obtained value from the three biological replicates is represented by a plot. Error bars of the graph are the SD. The statistically significant differences between wild-type and mutant strains were determined with a Mann–Whitney *U* test and the *P* value is indicated. (C) Comparative western blot showing RagB in wild-type (WT), PS2062 (*ponA::spec*) and BKE27300 as negative control (*ragB::erm*) strains. Strains were grown in LB medium until $OD_{600nm} = 0.5$ at 37 °C, crude extracts were loaded (2, 4, or 8 μl) and separated on 12.5% SDS-PAGE, transferred to a nitrocellulose membrane, and RagB was detected using anti-RagB antibodies. Data information: In (A, B), data are presented as the mean ± SD. ns not significant, *P < 0.05, **P < 0.01 (Tukey's test and Mann–Whitney *U* test, respectively). Source data are available online for this figure.

bacterial two-hybrid interaction assays (Fig. 4A). The interaction between RagB and GpsB, used as a positive control, (Cleverley et al, 2019) was confirmed, and we were able to detect an interaction between RodA and RagB but not between RodA and GpsB (Fig. 4A). To obtain additional evidence in support of this interaction, we next performed in vitro pull-down experiments. When purified recombinant full-length 6His-RagB was incubated with a crude extract of *B. subtilis* cells producing RodA-GFP, the two proteins co-purified on a Ni-NTA affinity column (Fig. 4B, lane 7 on the top and bottom panels). Non-specific binding of RodA-GFP to Ni-NTA resin was excluded by performing the same purification in the absence of 6His-RagB (Fig. 4B, lane 4, top membrane); in this condition, RodA is only present in the flow-through fraction (lane 2). To further confirm the interaction between RagB and RodA in the native environment, we carried out co-immunoprecipitation experiments (Fig. 4C). Crude extracts of *B. subtilis* cells expressing RodA-GFP in the wild-type (lane 3) and the Δ*ragB* (lane 2) backgrounds were purified on a resin carrying antibodies that specifically retain the GFP protein. RodA-GFP was retained on the resin and then eluted by disruption of the interaction with the antibodies (Fig. 4C, lanes 2 and 3, top membrane). In the elution fraction, we were able to reveal the presence of RagB with antibodies specifically directed against it (Fig. 4C, lane 3, bottom membrane). To exclude non-specific interactions of RagB with either the GFP or the resin, immuno-precipitation was also carried out using a crude extract of cells expressing soluble GFP. In this condition, GFP retained on the resin was eluted, but we did not detect RagB in the elution fraction (Fig. 4C, lane 1). Taken together, these results show an interaction between RagB and RodA both in vitro and in vivo.

## RagB may regulate RodA GT activity in the cell

The results of our interaction test suggest that RagB might regulate RodA activity through protein-protein interaction. In order to monitor the GT activity specifically related to SEDS proteins, experiments were carried out in a genetic background in which the genes encoding the 4 aPBPs *of B. subtilis* are deleted (strain AG157, Δ4) (Emami et al, 2017). In this genetic context, we placed a copy of *rodA* under control of a xylose-inducible promoter, in the presence or in the absence of *ragB* (strains SG1150 and SG1158, respectively). As previously described, the Δ4 strain was only able to grow in the presence of magnesium in the culture medium that limits the increased autolytic activity of this strain (Emami et al, 2017; Tesson et al, 2022) (Fig. 5A). The same growth profiles were observed for the other two strains Δ4 P$_{xyl}$*rodA* and Δ4 Δ*ragB* P$_{xyl}$*rodA* (Fig. 5A). In the presence of 0.5% of xylose, over-production of RodA was able to rescue the growth of strain SG1150 (Δ4 P$_{xyl}$*rodA*) but not that of strain SG1158 (Δ4 Δ*ragB* P$_{xyl}$*rodA*) (Fig. 5A). These results show that in the absence of RagB, overproduction of RodA is unable to compensate for the absence of the 4 aPBPs, suggesting that RagB stimulates the activity of RodA. This stimulation, dispensable in the presence of the aPBPs, becomes essential when SEDS proteins provide all GT activity in *B. subtilis* or when only PBP1 is absent, as previously suggested by growth complementation experiments (Fig. 2A,B).

We next investigated the morphology of these four strains (WT, Δ4, Δ4 P$_{xyl}$*rodA*, and Δ4 Δ*ragB* P$_{xyl}$*rodA*) during exponential growth in the presence of xylose (Fig. 5B,C). In total, 300 cells from three biological replicas each were analyzed. As mentioned above, cells of the Δ*ponA* mutant are thinner than wild-type cells, and

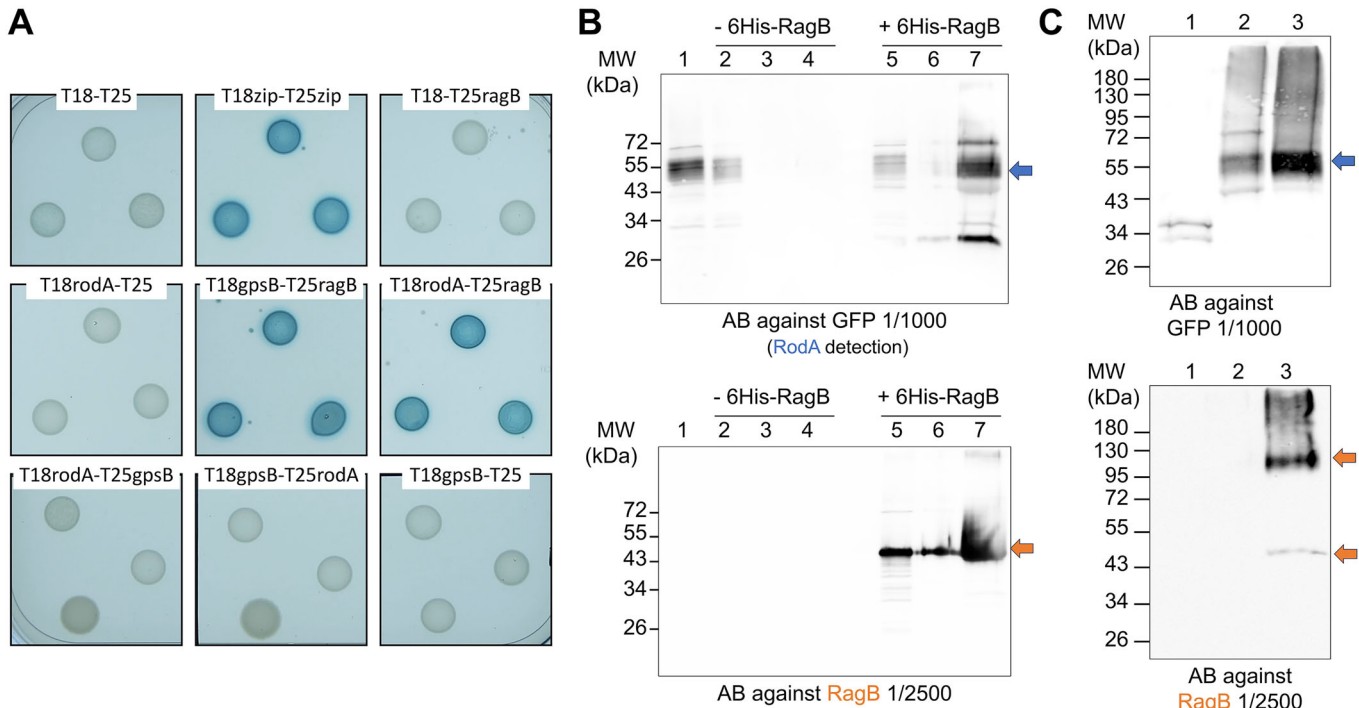

**Figure 4. RagB and RodA interact in vitro and in vivo.**

(A) Detection of protein interactions by bacterial two-hybrid assay. The T18 and T25 fragments of the adenyl cyclase protein were fused to the N-termini of RagB, GpsB, and RodA. Co-transformed strains of *E. coli* BTH101 were spotted onto LB medium supplemented with Xgal and IPTG and incubated at 30 °C overnight. Blue colonies indicate a positive interaction. Some were detected between GpsB and RagB, RodA and RagB, but no interaction was detected between GpsB and RodA. (B) Detection of protein interactions by pull-down assay. An extract of membrane proteins from the SG825 (*ragB::erm amyE::P*xyl*rodA-gfp*) strain containing RodA-GFP (lane 1) was incubated for 1 h at 4 °C with either buffer (lanes 2, 3, 4) or 100 μg of 6His-RagB protein (lanes 5, 6, 7) and then purified on a Ni-NTA column. The purification fractions were separated by SDS-PAGE and transferred to a nitrocellulose membrane. The presence of RagB (bottom gel) and RodA-GFP (top gel) proteins in the different fractions was detected using specific antibodies. In the protein extract (lane 1) RodA-GFP is detected confirming its production and extraction (top gel). In the non-retained fraction (lanes 2 and 5), some unbound RodA-GFP is detected, but not in wash fractions (lanes 3 and 6). During purification of 6His-RagB (bottom gel), an excess of protein is found in the non-retained (lane 5) and wash (lane 6) fractions, but the majority is eluted (lane 7). RodA-GFP is co-eluted (lane 7) with 6His-RagB. (C) Detection of protein interactions by Co-immunoprecipitation assay. Extracts of membrane proteins from the SG187 (*amyE::P*xyl*gfp*), SG818 (*amyE::P*xyl*rodA-gfp*) and SG825 (*ragB::erm amyE::P*xyl*rodA-gfp*) strains were purified on GFP-affinity resin and eluted proteins RagB (bottom gel) and GFP or RodA-GFP (top gel) were detected using anti-RagB or anti-GFP antibodies, respectively. No trace of non-specifically co-purified RagB is detected (lane 1, bottom gel) with GFP alone (lane 1, top gel) showing that RagB does not bind non-specifically to either GFP or the resin. RodA-GFP produced in SG825 and SG818 strains is well purified (lanes 2 and 3, top gel), and RagB from SG818 is specifically co-purified with RodA (lane 3, bottom gel) but not from SG825 (lane 2, bottom gel). Source data are available online for this figure.

ΔponA ΔragB mutant cells are not only thinner but also more curved (Fig. 1C). As expected, Δ4, Δ4 PxylrodA and Δ4 ΔragB PxylrodA cells in which all aPBPs (including PBP1) are absent, were also thinner than wild-type cells (Appendix Fig. S5). Furthermore, cell curvature also appeared to be modified when RodA was overproduced (Fig. 5B,C). We calculated the values for four morphological parameters related to cell curvature (longitudinal and lateral asymmetries, defined as the overlap ratio between the two halves of the cell contour; width variation defined as the standard deviation of the cell width and solidity, defined as the ratio of cell surface to convex cell surface) and all four showed statistically significant differences between these strains. Cells of the Δ4 mutant were more curved than wild-type cells, which results in an increase in longitudinal and lateral asymmetries, an increase in width variations and a decrease in solidity within a same cell (Fig. 5C). Interestingly, this curvature was significantly reduced when RodA was overproduced (Δ4 PxylrodA), with values similar to those of the wild-type strain for the four parameters. However, the effect of RodA overproduction on cell curvature was suppressed in

the absence of RagB, as observed in the strain Δ4 ΔragB PxylrodA for which the values of these parameters remained equivalent to that of Δ4 (Fig. 5C). These results indicate that in the absence of RagB, overproduction of RodA can no longer compensate for the morphology defects of the Δ4 strain, further showing that RodA activity may be dependent on RagB.

It is important to note that, despite several attempts, it was never possible to obtain the deletion of *ragB* in the Δ4 strain. This reinforces the hypothesis that RagB is necessary to stimulate (at least) RodA activity and, if it is absent, then there would be too little GT activity in the cell lacking aPBPs to allow it to grow.

## RagB forms randomly diffusing foci in the membrane

Although our results suggest a regulation of RodA activity by RagB, we cannot exclude that the defect in growth and morphology of the ΔragB ΔponA mutant might be explained by mislocalization of RodA in the absence of RagB, which would prevent its proper functioning in the Rod complex. To test this hypothesis, we first

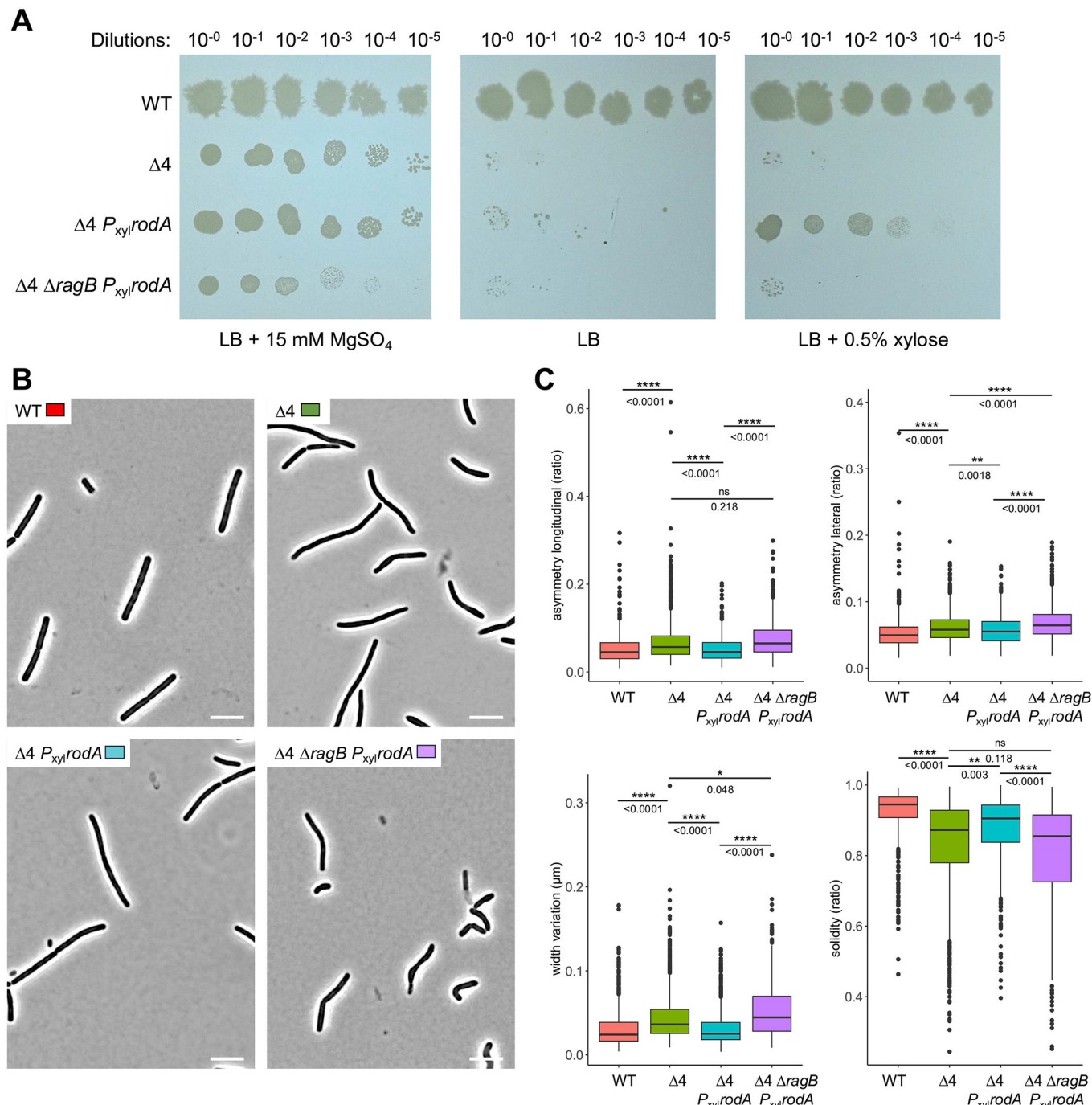

investigated the subcellular localization of RagB. Its structure predicted with Alphafold (https://alphafold.ebi.ac.uk/entry/O32031) displays a very short cytoplasmic domain, a transmembrane helix and an extracellular domain composed of a long-unfolded region, estimated by two softwares to be intrinsically disordered (IDR) (DISOPRED (Jones and Cozzetto, 2015) ; PrDOS (Ishida and Kinoshita, 2007)), and a structured C-terminal subdomain of unknown function (DUF1515) (Appendix Fig. S6). In addition, since RagB has been associated with proteins of the elongasome (Cleverley et al, 2019), its localization could be

dynamic. We constructed a strain expressing a *gfp-ragB* fusion under control of a xylose-inducible promoter, and we followed GFP-RagB localization during cell growth. The GFP-RagB fusion was functional since the Δ*ponA* Δ*ragB* strain producing this fusion protein had a growth similar to the Δ*ponA* strain (Fig. 6A). In exponentially growing cells, RagB localized to the cell membrane and was enriched at the septum (Fig. 6B), and in stationary phase cells, GFP-RagB was more present on the sidewalls (Fig. 6C). The fluorescent signal along the cell cylinder appeared discontinuous in epifluorescence imaging, suggesting that RagB is not evenly

**Figure 5. RodA overproduction rescues the growth of Δ4 strain only in the presence of RagB.**

(A) Serial dilutions of strains wild-type 168 (WT), AG157 (*pbpG::kan ΔpbpD ΔpbpF ΔponA* named Δ4), SG1150 (Δ4 *amyE*::P$_{xyl}$*rodA*) and SG1158 (Δ4 *ragB::erm amyE*::P$_{xyl}$*rodA*) grown in LB supplemented with 15 mM MgSO$_4$ until OD$_{600nm}$ = 0.4 were spotted (10 µl) on LB-Agar + 15 mM MgSO$_4$, LB-Agar and LB-Agar + 0.5% xylose plates and incubated overnight at 37 °C. Differences in growth between strains for both conditions are monitored by the appearance of colonies for each dilution. (B, C) Statistical analysis from microscopic morphology monitoring of Δ4 strain and derivatives. (B) Visible light microscopy images were taken for the 4 strains, wild-type 168 (WT), AG157 (*pbpG::kan ΔpbpD ΔpbpF ΔponA* named Δ4), SG1150 (Δ4 *amyE*::P$_{xyl}$*rodA*) and SG1158 (Δ4 *ragB::erm amyE*::P$_{xyl}$*rodA*) grown in LB supplemented with 2.5 mM MgSO$_4$ and 0.5% xylose until OD$_{600nm}$ of 0.4 at 37 °C. A representative image of the morphology of the cells of each strain is shown here. Ten images per strain were taken in order to be able to statistically process the morphology parameters in a significant number of cells. Data were collected from three biological replicates. Scale bar is 5 µm. (C) Statistical analysis of the values for four morphological parameters of cell shape representative of curvature measurement: lateral and longitudinal asymmetries (defined as the overlap ratio between the two halves of the cell contour separated by the median axis or the axis perpendicular to the middle of the cell, respectively, where 1 is the 100% of overlap), width variation (defined as the standard deviation of the cell width measured perpendicular to the median axis at a distance of one pixel apart) and solidity (defined as the ratio of cell surface to convex cell surface (convex hull) with 1 means that the cell shape is convex). Graphs represent the solidity values (0 to 1 ratio), the lateral and longitudinal asymmetry values (0 to 1 ratio) and the width variation along the main axis of the cell for each strain (µm). The statistical analysis of the whole data was obtained by a linear mixed-effects model (LMM) or a generalized linear mixed-effects model (GLMM). The contrast between strains was obtained from a global nonlinear model and shows significant differences; the exact *P* values of the two-by-two comparisons are indicated. Data information: In (C), data are presented as box plots. Each box represents the interquartile range (IQR), with the lower and upper bounds corresponding to the 25th and 75th percentiles, respectively. The center line indicates the median (50th percentile). Whiskers extend to the minimum and maximum values within 1.5 × IQR; data points beyond this range are considered outliers. ns = not significant, *$P < 0.05$, **$P < 0.01$, ****$P < 0.0001$ (global nonlinear model test). Source data are available online for this figure.

distributed in the membrane and may form discrete assemblies. Several proteins involved in CW synthesis in *B. subtilis*, including RodA, form distinct foci at the membrane with a subpopulation displaying a characteristic circumferential motion, supposedly reflecting the movement of the Rod complex (Domínguez-Escobar et al, 2011; Garner et al, 2011). To test if RagB may display such dynamics, we used Total Internal Reflection Fluorescence (TIRF) microscopy (Fig. 6D–F), a technique of choice for studying localization and dynamics of membrane-associated complexes (Cornilleau et al, 2020). In growing cells, RagB was not evenly distributed in the membrane, forming constantly reorganizing loosely defined clusters over the cell periphery (Fig. 6E; Movie EV1). Maximum projections and kymograph analysis failed to reveal obvious trajectories (Fig. 6F). This suggests that GFP-RagB does not display a directional movement in conditions of observation in which were previously observed such dynamics with proteins of the Rod complex (Domínguez-Escobar et al, 2011; Garner et al, 2011).

## RagB does not affect the localization or dynamics of RodA

A recent study on the dynamics of proteins of the Rod complex in *B. subtilis* suggested that RodA, PbpH and MreB molecules have similar trajectory orientations and velocities but different trajectory lengths (Dersch et al, 2020). Some RodA molecules are associated with the Rod complex and display directed circumferential motion, while some diffuse in the membrane (Dersch et al, 2020; Domínguez-Escobar et al, 2011). To check if RagB influences the localization or dynamics of RodA, we constructed a strain producing Halo-tagged RodA from the *rodA* locus under control of its native regulatory region (strain RCL1445). Because RodA is essential and its depletion leads to strong shape defects, rounding and lysis (Emami et al, 2017; Henriques et al, 1998), observation of the recombinant strain indicated that RodA functionality was unaffected by the tag, the strain presenting no shape defects and supporting wild-type growth (Fig. 7A,B). Observation of the fusion protein by time-lapse TIRF microscopy showed a discrete localization, highly dynamic, with many foci quickly diffusing in the membrane (Fig. 7C; Movie EV2), as previously reported with a natively produced RodA-GFP fusion (Domínguez-Escobar et al,

2011) and an inducible YFP-RodA fusion (Dersch et al, 2020). We explored a range of exposure times and frequencies of acquisition and found that a subpopulation of directionally moving foci could be visualized by kymograph analysis when using 1 image/s frequency and 0.9 s exposure time (Fig. 7D; Movie EV3). RodA moved processively in trajectories oriented perpendicularly to the long axis of the cells (Fig. 7D), as previously reported (Domínguez-Escobar et al, 2011). Using an automatic detection and tracking of particles formerly described (Billaudeau et al, 2017), we observed that RodA moved at a constant speed of $56.5 \pm 11.3$ nm/s, consistent with previously measured speeds and presumed to correspond to the speed of active Rod complexes (Domínguez-Escobar et al, 2011). The mean density of directionally moving foci of $0.128 \pm 0.077 \ \mu m^{-2}$, was lower than the one estimated for MreB ($\sim 0.4 \ \mu m^{-2}$) in *B. subtilis* (Billaudeau et al, 2017), albeit on the same order of magnitude. We next analyzed the speed and density of processive Halo-tagged RodA foci in the Δ*ponA*, Δ*ragB* and Δ*ragB* Δ*ponA* backgrounds. No significant differences were detected in the average RodA speeds in these three mutant backgrounds relative to the wild-type (Fig. 7E). However, we observed a slight reduction in the density of directionally moving RodA foci in strains without *ragB*. The average density of foci of the wild-type was $0.128 \pm 0.077 \ \mu m^{-2}$, versus $0.132 \pm 0.119 \ \mu m^{-2}$ in Δ*ponA*, $0.106 \pm 0.059 \ \mu m^{-2}$ in Δ*ragB* and $0.100 \pm 0.067 \ \mu m^{-2}$ in Δ*ragB* Δ*ponA*. Only the difference in density between the wild-type and the Δ*ragB* Δ*ponA* mutant ($\delta = 0.028 \ \mu m^{-2}$) was statistically significant, although that between the wild-type and the single Δ*ragB* mutant ($\delta = 0.022 \ \mu m^{-2}$) was similar (Fig. 7F). The density remained in a close range in the Δ*ponA* mutant to that in the wild-type (increase of $0.004 \ \mu m^{-2}$) (Fig. 7F), showing no effect of *ponA* deletion. These findings suggest that RagB has a limited impact on the localization and activity of RodA associated with the Rod complex.

We next wondered if RodA and RagB could display some activity outside of the Rod complex, similarly to aPBPs. In previous studies, analysis of the dynamic properties of aPBPs by the cumulative distribution function (CDF) approach revealed the existence of two subpopulations of diffusing particles, the slower subpopulation being proposed to actively synthesize PG (Cho et al, 2016; Vigouroux et al, 2020). We thus wondered if several

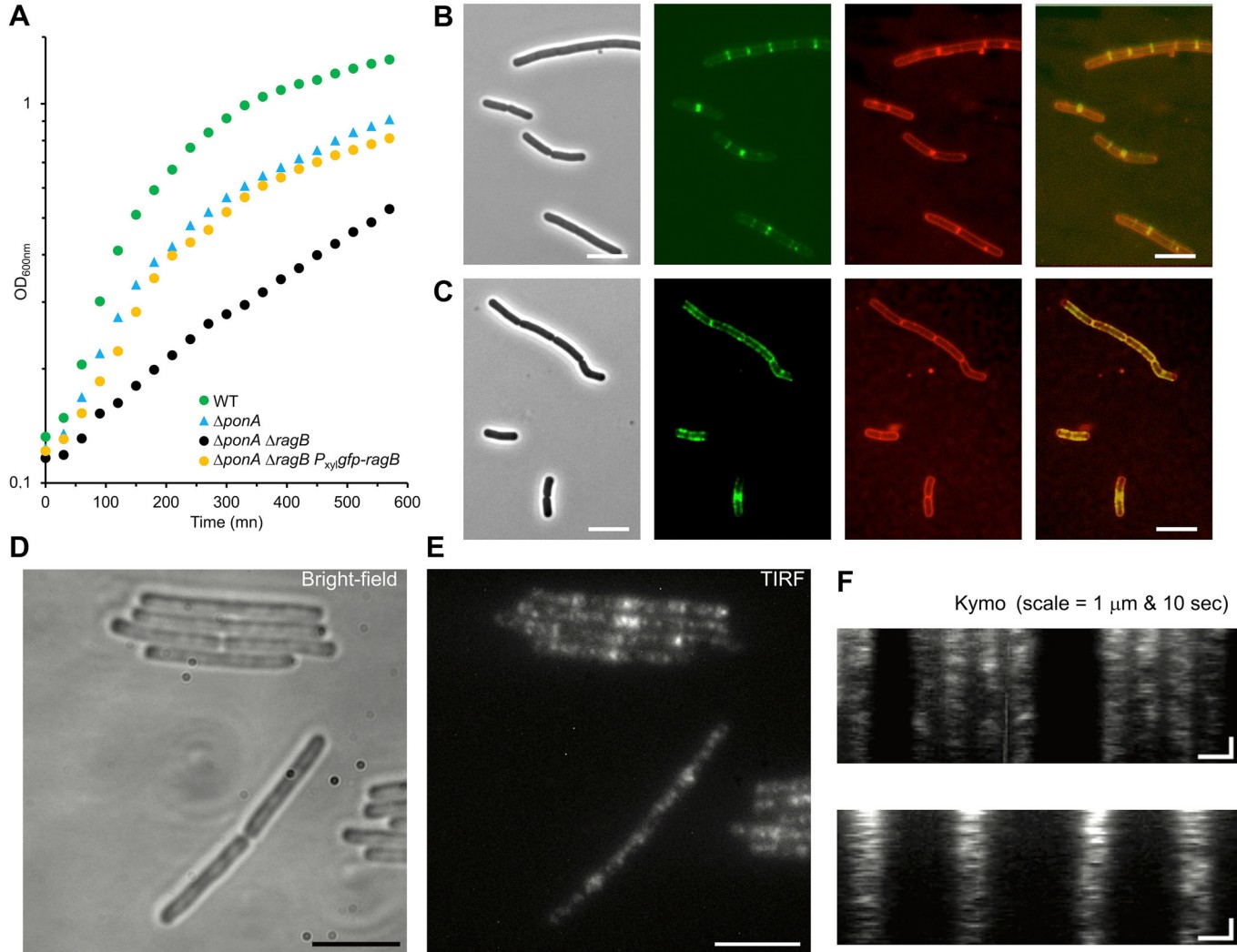

**Figure 6. Cellular localization of RagB during cell cycle.**

(A) The wild-type (WT, green) PS2062 (*ponA::spec*, blue), SG667 (*ponA::spec ragB::erm*, black) and SG1014 (*ponA::spec ragB::erm amyE::P_xyl gfp-ragB*, yellow) strains were inoculated from an exponentially grown culture at an $OD_{600nm}$ 0.1 in LB medium supplemented with 1% xylose in a 96-well plate in triplicate and grown until the stationary phase. The representative growth curves of three biological replicates are shown. (B, C) Strain SG661 (*ragB::erm, amyE::P_xyl gfp-ragB*) observed by phase contrast (left) or under epifluorescence illumination at 488 nm (middle in green) or 561 nm (middle in red) or a merged image (right). SG661 was grown in LB medium supplemented with 1% xylose at 37 °C, and GFP-RagB was localized by fluorescence microscopy (middle panels in green). For membrane labeling (middle panels in red), 1 ml of bacterial culture at $OD_{600nm} = 0.4$ (in exponential phase, (B) or at $OD_{600nm} = 3.7$ (in stationary phase, (C) was incubated for 1 min with 1 µl of 1 mg/ml FM4-64 reagent before preparing the slides. GFP-RagB is mainly observed at the cell membrane and enriched at the septum during the exponential phase and in the sidewalls during the stationary phase. (D, E) GFP-RagB (SG661; *ragB::erm, amyE::P_xyl gfp-ragB*) observed by TIRF microscopy epifluorescence (E) and under bright-field illumination (D). Scale bars are 5 µm. (F) Typical kymographs monitored on tracks drawn perpendicularly to the cell long axis, on a time-lapse acquisition by TIRFM of SG661. No specific trajectory is observed. Scale bars, 1 µm (horizontal) and 10 s (vertical). Source data are available online for this figure.

subpopulations of diffusing RodA could also be detected, and if the ratio between them would be altered in the absence of PBP1 or RagB. Due to the high density of RodA foci per cell which made conventional tracking unfeasible, we turned to single-particle tracking (SPT) using low concentration of Halo ligand, as described in the Materials and Methods section. The high signal-to-noise ratio and stability of the Halo ligand allowed us to characterize 4000–7000 individual trajectories per strain in four independent replicas. Unlike aPBPs, RodA displays a subpopulation of directionally moving foci that could alter the CDF analysis, and thus this population was excluded from the analysis. To achieve

this, we first classified particle behaviors using a mean square displacement (MSD) approach. This method, previously used to characterize MreB dynamics (Billaudeau et al, 2020; Billaudeau et al, 2017) revealed that 10–15% of RodA foci exhibit directional motion in both wild-type and Δ*ponA*, Δ*ragB* and Δ*ragB* Δ*ponA* mutant strains (Appendix Fig. S7A). These directionally moving foci were excluded from further analysis. Using the CDF approach, we next determined the diffusion coefficient (DC) of individual tracks and plot the distribution of DC per strain and replica, revealing bimodal distributions (Appendix Fig. S7B). The mean DC of each subpopulation and their ratio were directly estimated from

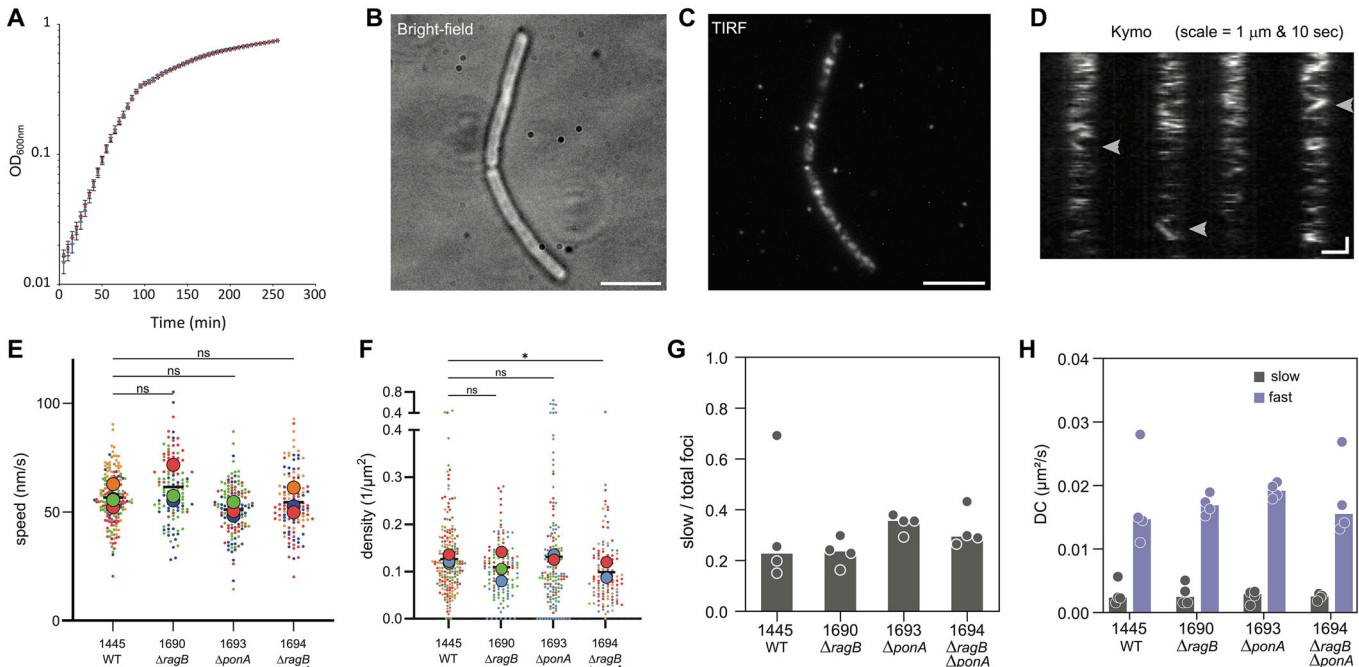

**Figure 7. Dynamic localization of RodA.**

(A) The wild-type 168 (blue) and RCL1445 (*rodA::rodA-halo*; red) strains were inoculated at an $OD_{600nm}$ of 0.01 in a 96-well plate in triplicate, from an exponentially grown culture, and were grown in LB medium until the stationary phase. Error bars are the standard deviation to the mean of technical replicas. (B, C) RCL1445 (*rodA::rodA-halo*) grown exponentially in LB medium and spotted on a 2% agarose-LB pad, observed in bright-field illumination (B) or in TIRFM (C). Cells display no shape defect (B) and fluorescent RodA shows a discrete distribution of localization in the membrane (C). Scale bars, 5 µm. (D) Typical kymographs observed on tracks drawn perpendicularly to the cell long axis, on a time-lapse acquisition by TIRFM of RCL1445. Scale bars, 1 µm (horizontal) and 10 s (vertical). Arrows indicate typical traces of moving foci. (E, F) Mean speed (E) and density (F) of directionally moving RodA-Halo subpopulation in wild-type (WT) or three mutant backgrounds: Δ*ragB* (RCL1690), Δ*ponA* (RCL1693) and Δ*ragB* Δ*ponA* (RCL1694). Plots are the sum of all values from at least 115 particles per strain, from three or four biological replicates, distinguished by colors. Large dots represent the mean for each replicate, black lines the mean of the pooled replicates. Nested *t* test on the non-pooled replicates were performed to analyze the differences between the means (speed or density) of the wild-type and each mutant. Only the mean density between the WT and the Δ*ragB* Δ*ponA* (RCL1694) mutant was significantly different (* = $P < 0.05$, $P$ value = 0.0482). The speed differences observed in E are not significant (ns). (G, H) SPT and CDF analysis of diffusing RodA-Halo particles in wild-type (WT) and three mutant backgrounds: Δ*ragB* (RCL1690), Δ*ponA* (RCL1693) and Δ*ragB* Δ*ponA* (RCL1694), grown on s-EZRDM, revealing two subpopulations. Solid bars are median values of four biological replicates (dots). (G) Ratio of slow diffusing particles over total trajectories. (H) Diffusion coefficient (DC) of the slow (gray) and fast (blue) diffusing subpopulations. Data information: In (E, F), data are presented as Means (for each replicate and for the pool of replicates). ns not significant, *$P < 0.05$ (Nested *t* test). Source data are available online for this figure.

the fit to the distributions (see "Methods"). This revealed the existence in the wild-type background of fast (~0.0147 µm²/s) and slow (~0.0024 µm²/s) diffusing subpopulations of RodA (Fig. 7H), the latter accounting for 23% of the total (Fig. 7G). However, in the *ponA* mutant, the fraction of slow diffusing particles displayed a 1.56-fold increase, supporting the hypothesis that RodA compensates for the absence of PBP1 by engaging in non-directional assembly of PG, outside of the Rod complex. The deletion of *ragB* alone, in a context in which it is not essential, had no impact on the diffusion coefficient and the proportion of slow diffusing particles. However, in the absence of PBP1, a condition where RagB becomes essential, we could observe a slight reduction of the fraction of slow diffusing RodA in the absence of RagB. This might bear witness of a role of RagB in the repair activity of RodA.

## Discussion

In this study, we present evidence showing that RagB interacts with the SEDS protein RodA and most likely increase its enzymatic activity. Several

PBP regulators have been identified (Galinier et al, 2023) but none has been reported for SEDS proteins so far. RagB is potentially the first one.

Both the Rod complex and aPBPs possess GT and TP activities. The prevailing model established that, contrary to what was initially envisaged, de novo synthesis of PG glycosidic units is carried out by the Rod complex (elongasome), while aPPBs provide fortifications and repairs to the pre-existing structure (Cho et al, 2016; Zhao et al, 2017). A recent model proposed that when PBP1 repairs PG damage, the IDR of this protein would be necessary to sense and locate the PG synthase to CW gaps to fortify them (Brunet et al, 2022). In *B. subtilis*, aPBPs are not essential though the Δ4 cells grow relatively slowly and require a high concentration of magnesium in the growth medium (McPherson and Popham, 2003), whereas RodA is essential (Emami et al, 2017). In addition, it was proposed that aPBPs are essential in bacteria, except where Rod complex components can replace them (Pazos and Vollmer, 2021). RodA GT activity, associated to the TP activity of the two co-essential bPBPs, PBP2A and PbpH, constitute the core of the elongasome and provide PG synthase activity in *B. subtilis* (Cho et al, 2016; Emami et al, 2017; Meeske et al, 2016).

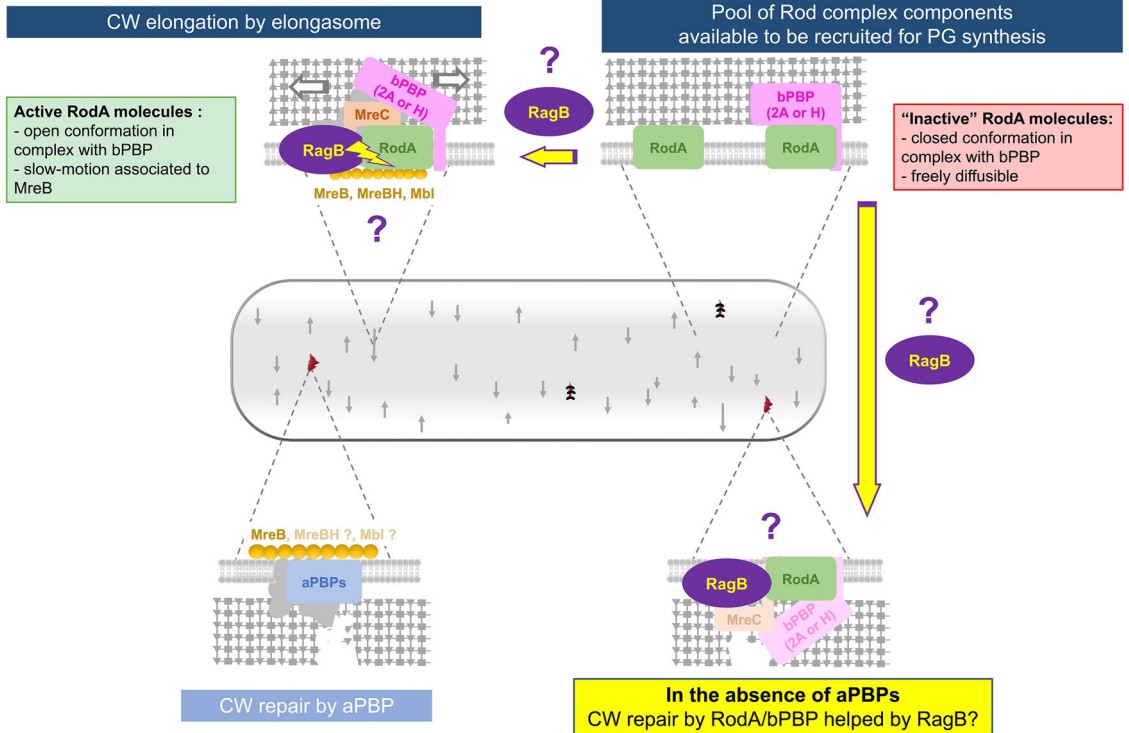

**Figure 8.   Model of the molecular role of RagB on the stimulation of RodA.**

In the presence of aPBPs, GT activities of RodA and PBP1 are sufficient for PG synthesis and repair. A stimulatory effect of RagB on RodA activity is dispensable. In the absence of aPBPs (or of PBP1), RodA assumes a fraction of the repairing activity while the elongasome components are upregulated (Patel et al, 2020) and RagB stimulatory effect on RodA activity becomes essential. Indeed, enough active Rod complex should take on these two functions (synthesis and repair). In this context, RagB may detect PG damages by its IDR and recruits freely diffusible molecules of RodA which, associated to its bPBP partners, will be able to repair them. RagB could also maintain an active conformation of RodA/bPBP complex to stimulate its activity outside or within the elongasome.

Here, we show that in the absence of the main aPBP, PBP1, the RagB protein becomes necessary for growth and normal morphology of the cells by enhancing PG synthase activity of RodA. Our data show that RagB does not appear to be involved in *rodA* expression nor in RodA protein levels but an interaction between RodA and RagB was observed. Since we detected free diffusion of RagB molecules rather than a directed circumferential motion, this suggests either a very transient interaction or that RagB interacts with RodA outside the elongasome. While the speed of RodA molecules associated to the Rod complex was not altered in the absence of RagB, we noticed a limited decrease in their density in *ragB* mutants. This difference seems specific to RagB since it was observed in both the ΔragB and ΔragB ΔponA strains but not in the single ΔponA mutant. We do not know how RagB boosts PG synthesis activity but based on our results and the current knowledge in the field, we can propose several avenues to explore.

One possibility is that RagB stimulates specifically RodA activity by modifying the dynamic of RodA molecules associated to the Rod complex, and thus PG synthesis activity of the complex (Dersch et al, 2020). Since our microscopy data do not reveal any change in the speed of the slow-moving population of RodA but a limited decrease in their density in the absence of RagB, it cannot be excluded that RagB might change the distribution of RodA molecules between those that diffuse freely and those associated with the elongasome. We can speculate that RagB could promote

association of freely diffusible RodA to the elongasome to increase the number of active Rod complexes or it could also help to maintain RodA within the elongasome. This stimulation of RodA by increasing its recruitment (or steadiness) within the elongasome complex could play an important role in particular in the absence of aPBP. This model is in agreement with the observation that RodA cellular levels or assembly to the elongasome would control the concentration of active Rod complex within the cell (Middlemiss et al, 2024).

RagB could also ameliorate interactions of RodA with other partners, resulting in enhanced GT activity. RodA acts in combination with its cognate bPBPs (PbpH and PBP2A in *B. subtilis*), the RodA/bPBP pair constituting the essential PG synthase having both GT and TP activities. A recent publication proposed a conserved mechanism necessary for the activation of PG synthesis during cell elongation (Shlosman et al, 2023). Using single-molecule FRET and cryo-EM, the authors observed that the complex composed of RodA and its bPBP partner dynamically switches between closed and open conformations. The RodA/bPBP complex is mainly in the inactive closed state, with enzymatic activity reduced to basal levels, which would effectively protect the cell from unwanted catalysis outside the Rod complex (Shlosman et al, 2023). The active open state connects the allosteric stimulation of polymerization and crosslinking. The cytoskeletal protein MreC likely contributes to PG synthesis through structural

dynamics of the bPBP in vivo and this MreC role is essential for this allosteric activation of enzymatic activity. A possible role of RagB could be to favor the switch between the closed and the active conformations of RodA/bPBP and/or to maintain it under its active conformation to stimulate its GT activity.

We could also imagine that RagB, which was reported to interact with GpsB, PBP1, PBPI, RodZ and YpbE (Cleverley et al, 2019) has a pleiotropic role. This hypothesis is also reinforced by the recent discovery of a genetic interaction between *ragB* and *ezrA* (Koo et al, 2024) which was confirmed by morphological defects in the deletion strains. RagB could therefore function as an adapter protein that mediates interactions and/or dynamics between PG enzymes and scaffolding proteins inside the Rod complex to regulate activity of elongasome at specific sites depending of the needs of the bacterial cell. A concomitant role for GpsB and RagB in CW synthesis is possible, but the additive effect of *ragB* and *gpsB* deletions in a Δ*ponA* mutant suggests that they are involved in different pathways too.

However, data on the localization of RagB showing freely diffusing molecules would not be entirely consistent with this model and we could also consider a second scenario in which RagB acts with RodA outside the Rod complex to conduct aPBP-like functions in cell wall repair. Several of our data support this model. Indeed, (i) RagB is necessary to boost RodA activity to rescue growth in the absence of PBP1; (ii) In the absence of RagB and PBP1, cells are slightly wider; and (iii) RagB has no effect on the dynamic of the Rod complex. Our data show that, similarly to what was reported for aPBPs (Billaudeau et al, 2020; Cho et al, 2016), the freely diffusing population of RodA is composed of two subpopulations with a large difference in their diffusion coefficients. By analogy with the model proposed for aPBPs, the slower diffusing RodA may reflect non-directional PG synthesis, i.e., a repairing activity, suggesting a redundancy between the two family of GT for this function. The increased fraction of slow diffusing RodA observed in the *ponA* mutant might bear witness of an increased need for disordered PG synthesis in the absence of PBP1 that RodA would compensate. In this model, RagB may serve to drag more RodA to the repairing pathway from the pool of inactive, fast diffusing ones.

Furthermore, the RagB predicted 3D-structure presents an extracellular IDR (Appendix Fig. S6) like in PBP1. This region is enriched in charged and polar amino acids, in particular K, S and D, which are also found in the IDR of PBP1 from *B. subtilis* or *B. cereus*. It was proposed that the IDR of PBP1, dispensable for its GT activity but necessary for its full cellular function, directs the enzyme to gaps in the damaged PG in order to repair them (Brunet et al, 2022). Here, in the absence of aPBP, we observed that RagB was crucial. We could hypothesize that the IDR of RagB plays a similar role than that of PBP1 and directs RodA to PG damage sites. In the absence of aPBPs, this RagB-dependent repairing activity by RodA would become essential, as we observe. Additional interaction tests between RodA and truncated versions of RagB showed that the transmembrane region of RagB and its IDR are necessary for the interaction with RodA (Appendix Fig. S8A–D), whereas the C-terminal domain is not. We also observed that RagB[15-120] protein deleted from its cytoplasmic N-terminal region (AA 1–15) and its extracellular C-terminal domain (DUF1515) is able to replace full-length protein during bacterial growth (Appendix Fig. S8E). This indicates that the two RagB domains interacting with RodA, *i.e.* the transmembrane region and the IDR,

appear to be sufficient for its function in vivo. These observations support the hypothesis that RagB IDR could be involved in this RodA-booster function.

In summary, using all the data collected during this study and current knowledge of PG synthesis, we propose a regulatory model (Fig. 8). When both PBP1 and RodA are present, their PG synthase activities are sufficient for ordered PG synthesis and repair; the role of RagB is negligible and a deletion of *ragB* has no effect. The strain is sensitive to moenomycin. In the absence of aPBPs or of PBP1 alone, RodA must compensate for aPBPs missing repairing activity while maintaining ordered PG synthesis in the Rod complex. In this condition, a hypothesis would be that RagB might help dragging more RodA, in the absence of PBP1, to the pool of repairing RodA. In this context, RodA is upregulated following the cellular response signaling some damages to the cell wall, via σ[M] (Patel et al, 2020). The strain becomes resistant to moenomycin. However, increase of RodA activity in the absence of aPBPs requires both the upregulation via σ[M] and the presence of RagB which becomes essential. We propose several modes of action for RagB. It could target RodA to the sites of PG repair via its IDR and promote the association of freely diffusible RodA molecules to these damage sites. It could also endorse the association of freely diffusible RodA molecules to the elongasome or maintain an active conformation of RodA/bPBP complex to stimulate its activity outside or within the elongasome. Testing the model and the different proposed assumptions will require further investigations.

# Methods

### Reagents and tools table

| Reagent/resource | Reference or source | Identifier or catalog number |
|---|---|---|
| **Experimental models** | | |
| *Bacillus subtilis* WT168 | Lab collection | *trpC2* |
| PS2062 | Popham and Setlow, 1996 | *trpC2 ponA::spec* |
| JR46 | Tavares et al, 2008 | *trpC2 gpsB::kan* |
| BKE27300 | Koo et al, 2017 | *trpC2 ragB::erm* |
| BKE23000 | Koo et al, 2017 | *trpC2 ypbE::erm* |
| AG157 | Emami et al, 2017 | *trpC2 pbpG::kan ΔpbpD ΔpbpF ΔponA* |
| SG1150 | This work | *trpC2 pbpG::kan ΔpbpD ΔpbpF ΔponA amyE::P<sub>xyl</sub>rodA* |
| SG1158 | This work | *trpC2 pbpG::kan ΔpbpD ΔpbpF ΔponA ragB::erm amyE::P<sub>xyl</sub>rodA* |
| SG314 | This work | *trpC2 ponA::spec gpsB::kan* |
| SG955 | This work | *trpC2 ponA::spec gpsB::kan amyE::P<sub>xyl</sub>rodA* |
| SG667 | This work | *trpC2 ponA::spec ragB::erm* |
| SG865 | This work | *trpC2 ponA::spec ypbE::erm* |
| SG712 | This work | *trpC2 ponA::spec ragB::erm amyE::P<sub>xyl</sub>ragB* |
| SG713 | This work | *trpC2 ponA::spec ragB::erm amyE::P<sub>xyl</sub>rodA* |

| Reagent/resource | Reference or source | Identifier or catalog number |
|---|---|---|
| SG933 | This work | *trpC2 ponA::spec ragB::erm amyE::P$_{xyl}$rodA(D280A)* |
| SG818 | This work | *trpC2 amyE::P $_{xyl}$rodA-gfp* |
| SG825 | This work | *trpC2 ragB::erm amyE::P$_{xyl}$rodA-gfp* |
| SG661 | This work | *trpC2 ragB::erm amyE::P$_{xyl}$gfp-ragB* |
| SG1014 | This work | *trpC2 ponA::spec ragB::erm amyE::P$_{xyl}$gfp-ragB* |
| SG187 | This work | *trpC2 amyE::P$_{xyl}$gfp* |
| RCL1445 | This work | *trpC2, rodA::(rodA-Halo, bleo)* |
| RCL1676 | This work | *trpC2, ΔragB::kan* |
| RCL1690 | This work | *trpC2, rodA::(rodA-Halo, bleo), ΔragB::kan* |
| RCL1552 | This work | *trpC2, ΔponA::spec* |
| RCL1693 | This work | *trpC2, rodA::(rodA-Halo, bleo), ΔponA::spec* |
| RCL1680 | This work | *trpC2, ΔragB::kan, ΔponA::spec* |
| RCL1694 | This work | *trpC2, rodA::(rodA-Halo, bleo) ΔragB::kan, ΔponA::spec* |
| *Escherichia coli BTH101* | Battesti and Bouveret, 2012 | |
| *Escherichia coli BL21(DE3)* | Tagourti et al, 2008 | |
| **Recombinant DNA** | | |
| pET21a-*ragB* | This work | |
| pJPR1-*ragB* | This work | |
| pJPR1-*rodA* | This work | |
| pJPR1-*rodA(D280A)* | This work | |
| pSG1154-*rodA* | This work | |
| pSG1729-*ragB* | This work | |
| pT25-*ragB* | This work | |
| pT25-*gpsB* | Pompeo et al, 2015 | |
| pT25-*rodA* | This work | |
| pT18-*gpsB* | Pompeo et al, 2015 | |
| pT18-*rodA* | This work | |
| **Antibodies** | | |
| Rabbit anti-RagB | This work | |
| Rabbit anti-GFP | This work | |
| Goat anti-Rabbit | Thermo scientific | 32460 |
| **Oligonucleotides and other sequence-based reagents** | | |
| Primer cc482 | This work—3′ *rodA* | TGGGAACACGGATT GGATTCAG |
| Primer cc483 | This work—3′ *rodA* | CCAGGACCTTGTCCGCTAC CCTCAAGAGAATTAAAAA GATAACTTCTGTATTTCGTCAG |
| Primer cc484 | This work— downstream *rodA* | TCAACAAATAAAAGCT AAAATCTATTATTAATCTG TTCAGCAATCTATGCAGACA GCCTTTACAGAGG |

| Reagent/resource | Reference or source | Identifier or catalog number |
|---|---|---|
| Primer cc485 | This work— downstream *rodA* | AAAGCCGCTCACACTGCTTATC |
| Primer cc461 | This work—halo | CTTGAGGGTAGCGGACAAGG TCCTGG |
| Primer cc462 | This work—halo | ATTCTCACGCATAAAA TCCCCTTTCATTTTCTAA TGCGATTAGCCGCTGATTT CTAAGGT |
| Primer cc447 | This work—bleo | CATTAGAAAATGAAAGG GGATTTTATGCGTGAGAAT |
| Primer cc448 | This work— bleo | GATTGCTGAACAGAT TAATAATAGATTTT AGCTTTTTATTTGTTGA |
| Primer pET21a-ragB-f | This work | GCAGGATCCAGCAATA ATCAATCTCG |
| Primer pET21a-ragB-r | This work | TTTCTCGAGTTTTAGCT TTTCTACTTTTGTCGG |
| Primer pSG1729-ragB-f | This work | GCACGGATCCTGAGCAATA ATCAATCTCG |
| Primer pSG1729-ragB-r | This work | TGCAAGCTTTTATTTTAGC TTTTCTACTTTTGTCGG |
| Primer pSG1154-rodA-f | This work | ATGGGTACCAGTCGATATA AGAAACAGCAAAGCCCC |
| Primer pSG1154-rodA-r | This work | TCGGAATTCAGAATTAA AAAGATAACTTCTGTATTTCG |
| Primer pJPR1-rodA-f | This work | ATGTCTAGATGAAAATGTAA GGCGGGATAGAATGAG |
| Primer pJPR1-rodA-r | This work | GATGCGGCCGCAGCCTCTGTA AAGGCTGTCTGC |
| Primer rodA(D280A)-f | This work | CCAGAGAGTACGACTGCCT TTATCTTTTCTATA |
| Primer rodA(D280A)-r | This work | TATAGAAAAGATAAA GGCAGTCGTACTCTCTGG |
| Primer pJPR1-ragB-f | This work | GAGTCTAGAAGGAGTGCAG AATTAATGAGCAAT |
| Primer pJPR1-ragB-r | This work | AAAGCGGCCGCTTTT AACCCGGCTGCTTTTGTTTA |
| Primer T25-ragB-f | This work | GCAGCTGCAGGGAGCAA TAATCAATCTCGTTAT |
| Primer T25-ragB-r | This work | GGCTGGATCCTCTTATT TTAGCTTTTCTACTTT |
| Primer T25-rodA-f | This work | GATAGGTACCCGTCA AGAATTAAAAAGATAAC |
| Primer T25-rodA-r | This work | CTAGAGGATCCCAGTCG ATATAAGAAACAGC |
| Primer T18-rodA-f | This work | GGGGTACCGATGAG TCGATATAAGAAACAGC |
| Primer T18-rodA-r | This work | TATCAAGCTTATAGA ATTAAAAAGATAACTTCT |
| **Chemicals, enzymes, and other reagents** | | |
| Standard antibiotics | Sigma | |
| Moenomycin | Euromedex | |
| LB medium | Gibco | |

| Reagent/resource | Reference or source | Identifier or catalog number |
|---|---|---|
| s-EZRDM | Barns and Weisshaar, 2013 | |
| FM4-64 | Molecular Probe | |
| Restriction enzymes and polymerases | NEB | |
| HaloTag ligand JFX-549 | Promega | |
| Superscript II Reverse Transcriptase | Invitrogen | |
| SYBR Premix Ex Taq | Takara Bio Group | |
| Ni-NTA resin | Qiagen | |
| Anti-GFP antibody resin | Chromotek | |
| Nitrocellulose membrane | Amersham | |
| ECL advanced reagents | GE Healthcare | |
| **Software** | | |
| MicrobeJ plug-in | ImageJ | Ducret et al, 2016 |
| Matlab- R2023b | Mathworks | |
| Matlab- R2024b | Mathworks | https://github.com/CyrilleBillaudeau/DiffusionQuantification_CDF |
| TrackMate | Tinevez et al, 2017 | |
| MetaMorph | Molecular Devices | |
| Ilas2 | Roper Sc. | |
| DISOPRED | Jones and Cozzetto, 2015 | |
| PrDOS | Ishida and Kinoshita, 2007 | |
| **Other** | | |
| Microplate reader | TECAN | Spark |
| Upright Axio Imager M2 Microscope | Zeiss | |
| Ti-E microscope | Nikon | |
| Elyra PS1 microscope | Zeiss | |
| Mastercycler | Eppendorf | Realplex |
| ImageQuant LAS4000 | GE Healthcare | |

## Plasmids and strains constructions

Standard procedures for molecular cloning and cell transformation were used. All strains, plasmids and primers used in this study are listed in the Reagents and Tools Table. All the PCR-derived DNA fragments in the plasmid constructs were verified by sequencing (Eurofins Genomics). To produce RagB with a C-terminal 6His-tag,

the *ragB* gene was amplified by PCR from chromosomal DNA of *B. subtilis* 168 using specific primers and introduced into the pET21a (+) plasmid (Novagen) between the BamHI and XhoI sites. The obtained recombinant plasmid was introduced in *E. coli* BL21 DE3 (Tagourti et al, 2008). For the generation of fusion proteins for the adenylate cyclase-based two-hybrid assays, *ragB, rodA* and *gpsB* genes were amplified by PCR from *B. subtilis* 168 chromosomal DNA using specific primers and inserted between PstI and BamHI sites for *ragB*, in BamHI site for *gpsB* (Pompeo et al, 2015) and between KpnI and BamHI sites for *rodA* in pT25. *rodA and gpsB* (Pompeo et al, 2015) genes were amplified by PCR inserted between KpnI and HindIII or KpnI and XhoI sites, respectively, in pT18. To overproduce RodA and RagB in several *B. subtilis* strains by xylose induction, *rodA* and *ragB* genes were amplified by PCR from *B. subtilis* 168 chromosomal DNA using specific primers and inserted between XbaI and NotI sites in pJPR1 plasmid. The obtained recombinant plasmids were introduced into the *amyE* locus of the *B. subtilis* suitable strains and selected for chloramphenicol resistance and a negative starch degradation test. Using the pJPR1-*rodA* plasmid, the point mutation D280A was introduced into the *rodA* gene by PCR amplification of the whole plasmid with a pair of primers designed with mismatching nucleotides at the center of the primers and containing the mutation. Then, the PCR products were incubated at 37 °C for 3 h with 1 µl of DpnI that digests methylated parental strands and transformed into *E. coli* DH5α. The resulting construct pJPR1-*rodA(D280A)* was verified by DNA sequencing. To produce xylose-inducible GFP fusion proteins, *rodA* and *ragB* genes were amplified by PCR from *B. subtilis* 168 chromosomal DNA using specific primers and inserted between KpnI and EcoRI sites in pSG1154 plasmid and between BamHI and HindIII sites in pSG1729 plasmid, respectively. The obtained recombinant plasmids were introduced at the *amyE* locus of the *B. subtilis* suitable strains and selected for spectinomycin resistance and a negative starch degradation test.

For the introduction of a *rodA-Halo* fusion at the native *rodA* locus, two DNA fragments carrying most of the 3′ end of the *rodA* open reading frame or the region downstream of *rodA* were PCR amplified from chromosomal DNA of wild-type *B. subtilis* 168 using oligonucleotides cc482/483 and cc484/485, respectively. Two additional PCR fragments carrying *halo* (encoding the Halo protein) and *bleo* (encoding a Phleomycin/Bleomycin resistance cassette), separately amplified using primers cc461/462 and cc447/448, respectively, were sandwiched between the two previous fragments and joined by long-flanking homology PCR and the external cc482/485 oligonucleotides. The resulting DNA fragment containing the 3′ fragment of *rodA* in translational fusion with *halo* followed by *bleo* and the downstream region of *rodA*, was transformed into wild-type 168 *B. subtilis* and selected for phleomycin resistance. The entire recombinant locus was checked by Sanger sequencing. The resulting construct (RCL1445; *rodA::rodA-Halo*, *phleo*) was subsequently combined with either the deletion of *ragB* (RCL1676; *ragB::km*) or *ponA* (RCL1552; *ponA::spec*) or both (RCL1680), generating strains RCL1690, RCL1693 and RCL1694, respectively, as summarized in the Reagents and Tools Table.

## General growth conditions

Luria-Bertani (LB) broth was routinely used for bacterial growth at 37 °C. For SPT, LB was replaced with s-EZRDM, a defined rich

medium with minimal auto-fluorescence (Barns and Weisshaar, 2013). When necessary, the appropriate antibiotics (Sigma) were added (ampicillin at 100 µg/ml, chloramphenicol at 50 µg/ml for *E. coli*, kanamycin at 10 µg/ml, spectinomycin at 100 µg/ml, erythromycin at 2 µg/ml, phleomycin at 15 µg/ml and chloramphenicol at 5 µg/ml for *B. subtilis*). When necessary, $MgSO_4$ at 15 mM is added in LB medium. LB-Agar plates are used for growth on solid medium supplemented with 15 mM $MgSO_4$ or 0.5% xylose when needed. To monitor the growth of the different strains tested, adequate volume of a 1 ml pre-culture in LB supplemented with 15 mM $MgSO_4$ was used to inoculate 150 µl of LB or LB + 15 mM $MgSO_4$ or LB + 1% xylose in triplicate on a 96-well plate to reach initial OD = 0.1. The plate was then incubated at 37 °C in a microplate reader (Spark, TECAN), with stirring, and $OD_{600nm}$ measurements were taken every 30 min for 7 h.

## Microscopy

Images were taken with a Zeiss Upright Axio Imager M2 Microscope (100X magnification of objective lens). 2 µl samples of bacterial cultures were loaded on the surface of a square of 2% LB-Agarose pad which was placed on a slide. A cover glass and a drop of oil were added before image acquisition. For GFP fluorescence, samples were exposed for 800 ms at 488 nm. For membrane labeling, 1 ml of bacterial culture was incubated for 1 min with 1 µl of FM4-64 reagent (Molecular Probe) before preparing the slides. Samples were exposed for 300 ms at 510 nm. For morphology analysis, 1 ml of bacterial culture at $OD_{600nm}$ 0.4, in LB supplemented with 2.5 mM $MgSO_4$ and 0.5% xylose at 37 °C was centrifuged for 3 min at 1000× *g* and resuspended in 100 µl of LB. In all, 2 µl samples were loaded on the surface of a square of 2% LB-Agarose pad placed on a slide, covered with a cover glass and a drop of oil before image acquisition for 20 ms. Data were collected from ten images for each strain and for each of the three replicas of the experiment and processed in order to compare the morphological descriptors in the different strains. The analysis workflow includes three main steps: segmentation, quantification and statistical description. Segmentation was obtained with a pre-trained deep learning model (Cutler et al, 2022). Quantification of morphological parameters was performed with a specific ImageJ/FIJI plug-in for microbiology MicrobeJ (Ducret et al, 2016; Schindelin et al, 2012). Statistical description was performed with R software (https://www.R-project.org/) and lme4 package (Bates et al, 2015). The statistical analysis of the whole data was obtained by linear mixed-effects model (LMM) or generalized linear mixed-effects model (GLMM).

## TIRF microscopy and analysis

TIRFM acquisitions were performed on an inverted Nikon Ti-E or a Zeiss Elyra PS1 microscope equipped with an EMCCD (electron-multiplying charge-coupled device) camera (iXON3 DU-897; Andor). The Ti-E carried an azimuthal TIRF, an iLas2 laser coupling system (Roper Scientific) (150 mW, 488 nm), a ×100 Apochromat TIRF oil objective (NA, 1.49; Nikon), a ×2.5 magnification lens and was piloted by MetaMorph (Molecular Devices) and Ilas2 (Roper Sc.). The PS1, controlled by Zen black (Zeiss), included a 100 mW laser (488 nm), a ×1.6 magnification lens and a 100× Apochromat TIRF oil objective (NA, 1.46; Zeiss). Cell growth, cell preparation and setting up of slides were described elsewhere (Cornilleau et al, 2020). Specifically, for Halo labeling,

cells at mid-log phase of growth ($OD_{600\ nm}$ 0.15–0.25) were sampled and mixed with HaloTag ligand JFX-549 (Janelia Fluor HaloTag, Promega) at a final concentration of 10 nM and incubated an extra 15 min with strong aeration before mounting on a 2% agarose-LB or agarose-s-EZRDM pad on a plasma-cleaned microscope slide. For SPT, the ligand concentration was reduced to the final concentration of 0.2 nM. Imaging parameters (exposition time, acquisition rate and length) encompassed a large range of conditions, and settings retained for the presented snapshots, videos and analyses are indicated in the corresponding legends. For SPT, time-lapse images were acquired continuously, with an exposure time of 100 ms and 1000 individual frames captured in total. Image processing, analysis and quantification of fluorescent particle dynamic were performed as previously described (Billaudeau et al, 2020; Billaudeau et al, 2017). Quantifications on RodA-Halo foci were performed mainly as previously described for GFP-MreB on similarly acquired TIRF images (Billaudeau et al, 2017). In short, individual particles were detected on each image of time series using a comet detection approach, followed by tracking between frames, with U-track (Matlab), and mean square displacement (Billaudeau et al, 2020) analysis to derive speed, directionality and densities (Matlab- R2023b; Mathworks).

## Quantification of fast and slow diffusing RodA particle

The analysis of the behavior of diffusive populations of RodA-HaloTag proteins is based on the extraction of the diffusion coefficient using the cumulative distribution function (CDF) approach. The data was processed through five distinct steps: (1) bacterial cells were segmented from the bright-field microscopy image; (2) the TIRF image sequences were pre-processed to increase the signal-to-noise ratio (by subtracting from the images the minimum projection of the stack intensity followed by time averaging by a sliding window over two consecutive images); (3) single-particle tracking was conducted with TrackMate (Tinevez et al, 2017) (detection was done using LoG filter set with radius = 0.25 µm and threshold = 100; reconnection with the LAP connector with linking max distance = 0.4 µm and max frame gap = 0); (4) non-directed trajectories were identified using the previously published MSD approach (Billaudeau et al, 2020; Billaudeau et al, 2017) (minimum trajectory duration: 10 frames; R2dir = 0.9; R2diff = 0.8); and (5) the quantification of diffusion coefficient of non-directed trajectories was performed using the CDF approach. The CDF analysis was performed on each trajectory individually (with a minimum duration of ten frames) using a one-component model:

$$CDF(r, \Delta t) = 1 - \exp\left(-\frac{r^2}{4D\Delta t}\right)$$

where CDF is the cumulative probability of a displacement of magnitude *r* given the diffusion coefficient *D* and the observation period $\Delta t = 4*t_{lag}$ ($t_{lag}$ is the interval time between two consecutive images). The diffusion coefficients were then compiled for all acquisitions from the same replica ($N_{replicat} = 4$). A thorough analysis of the distribution of diffusion coefficients was conducted, revealing bimodal behavior across all replica and biological conditions (Appendix Fig. S7B). This finding indicates the presence of two diffusion coefficients ($D_1$ and $D_2$) and a ratio $w_1$ (for the population 1),

which have been quantified using maximum likelihood estimates with a model that employs a mixture of two distinct log-normal distributions. The scripts used in this analysis have been developed in two distinct programming environments: the first three steps have been written in Fiji macros, while the fourth and fifth steps have been implemented in Matlab R2024b. These scripts are available on GitHub (https://github.com/CyrilleBillaudeau/DiffusionQuantification_CDF).

## Moenomycin resistance test

Strains were grown at 37 °C on LB medium supplemented with 15 mM MgSO₄ to an $OD_{600nm}$ of 0.4. A 800 µl aliquot of each culture was centrifuged for 3 min at 7000 rpm. Cell pellets were resuspended in 300 µl of LB and 200 µl were used to inoculate a 1% xylose LB-Agar plate that was dried for 20 min. Then filter paper disks containing 40 µg of moenomycin (Euromedex) were placed on the plates that were incubated at 30 °C overnight before measurement of the inhibition zones.

## Quantitative RT-PCR

In total, 1 µg of RNA was reverse-transcribed using the standard protocol of Superscript II Reverse Transcriptase (Invitrogen, USA) and 100 ng of random primers. The resulting cDNA was diluted 16-fold and 5 µl were used for the q-RT-PCR reaction. This step was performed on a Mastercycler® ep *Realplex* (Eppendorf) using the SYBR Premix Ex Taq (Perfect Real Time) PCR Kit (Takara Bio Group, Japan) in a final volume of 20 µl according to the manufacturer's instructions. Melting curves were analyzed to control for the specificity of the PCR reactions. Data from three biological replicates were analyzed and normalized with the software supplied with the Mastercycler®. Relative units were calculated from a standard curve plotting four different dilutions (1/80, 1/400, 1/2000 and 1/10,000) against the PCR cycle number (Ct) at which the measured fluorescence intensity reached the threshold, specified so that it is significantly above the noise band of the baseline (tenfold above the standard deviation value).

## Bacterial two-hybrid assays

The RagB, RodA, and GpsB proteins were fused to the T18 or T25 catalytic domain of adenylate cyclase using plasmids pT18 and pT25. Co-transformed strains of *E. coli* BTH101 carrying pT18-derivative plasmids and pT25-derivative plasmids were plated on LB-Agar and incubated at 30 °C for 48 h using the same protocol described in (Battesti and Bouveret, 2012). The strain carrying the pT18 and pT25 plasmids without inserts was used as a negative control. One milliliter of LB medium supplemented with 100 µg/ml ampicillin, 50 µg/ml chloramphenicol and 0.5 mM IPTG was inoculated and incubated overnight at 30 °C. Ten microliters of the overnight culture were spotted on the LB-Agar medium plates containing appropriate antibiotics, 0.5 mM IPTG and 40 µg/ml X-Gal. The plates were incubated overnight at 30 °C.

## Protein purification

Plasmid pET21a-*ragB* (Reagents and Tools Table) overproducing 6His-tagged protein RagB was transformed into *E. coli* BL21 DE3 (Tagourti et al, 2008). Purification of 6His-RagB was performed

with Ni-NTA resin (Qiagen) as previously described (Luciano et al, 2009) except that the proteins extraction buffer contains 0.5% 3-[(3-cholamidopropyl) dimethylammonio]-1-propanesulfonate (CHAPS, Sigma) and the purification buffers contain 0.05% CHAPS. The purified protein was stocked at −80 °C in a buffer containing 50 mM Tris-HCl, pH 7.5, 50 mM NaCl, 15% glycerol.

## Co-immunoprecipitation

For crude membrane preparation, SG187 (*amyE*::P$_{xyl}$*gfp*), SG818 (*amyE*::P$_{xyl}$*rodA-gfp*) or SG825 (*ragB::erm amyE*::P$_{xyl}$*rodA-gfp*) strains were grown in 100 ml of LB supplemented with 1% xylose at 37 °C. At $OD_{600\ nm} = 1$, 0.5% of formaldehyde was added for 15 min then cells were harvested for membrane preparation. Membrane proteins were solubilized by the addition of the nonionic detergent n-dodecyl-β-d-maltopyranoside(DDM, Sigma) to a final concentration of 0.5%. The soluble fraction was mixed with 20 µl anti-GFP antibody resin (Chromotek) and rotated for 2 h at room temperature. The resin was pelleted at 3000 rpm and the supernatant was removed. After washing, immunoprecipitated proteins were eluted by the addition of 60 µl of sodium dodecyl sulfate (SDS) sample buffer and heated for 15 min at 55 °C. The eluted material was transferred to a fresh tube, and 2-mercaptoethanol was added to a final concentration of 10%. Eluted proteins were separated by SDS-PAGE and visualized by western blot.

## Pull-down

Membrane proteins containing RodA-GFP were extracted from a culture of *B. subtilis* SG825 (*ragB::erm amyE*::P$_{xyl}$*rodA-gfp*) in LB supplemented with 1% xylose at 37 °C until $OD_{600}$ of 1 and resuspended in 50 mM HEPES pH 8, 200 mM NaCl, 1 mM MgCl₂, 1 mM CaCl₂ and 0.5% DDM. Proteins were incubated without or with 100 µg of purified 6His-RagB for 1 h at 4 °C then purified on Ni-NTA resin as previously described (Luciano et al, 2009) except that the buffers contained 0.1% DDM. Eluted proteins were separated by SDS-PAGE and visualized by western blot.

## Western blot

Samples were run on a 12.5% SDS-PAGE and transferred to a nitrocellulose membrane (Amersham) by electroblotting. The membrane was blocked with PBS-Tween 0.05% (PBST) solution containing 5% milk powder (w/v), for 3 h at room temperature with shaking, then incubated overnight at 4 °C with anti-RagB, or anti-GFP antibodies diluted to 1/2500th or 1/1000th, respectively. After three washes in PBST, the membrane was incubated for 1 h with the secondary peroxidase-conjugated Goat anti-Rabbit (Thermo Scientific) antibody, used at 1/2000th dilution in PBST. After three washes in PBST, the membrane was incubated with ECL advanced reagents (GE Healthcare) and scanned for chemiluminescence with an ImageQuant LAS4000 (GE Healthcare).

## Quantitative western blot

Strains were grown in LB medium until $OD_{600nm} = 0.5$ at 37 °C then centrifuged for 10 min at 8000 rpm at 4 °C. Cell pellets were resuspended in 1/200$^e$ volume of lysis buffer containing 10 mM

Tris-HCl pH 8.0, 150 mM NaCl, 0.1% NP40, 1 mM PMSF, 25 U/ml benzonase and 10 mg/ml lysozyme and incubated for 30 min at 37 °C then heated at 100 °C for 10 min after addition of 5X Laemmli buffer. Crude extracts were then loaded (2, 4, 8, or 16 μl) and separated on 12.5% SDS-PAGE, transferred to a nitrocellulose membrane treated as previously described. RagB was estimated with anti-RagB antibodies diluted to 1/2500th.

## Data availability

No large primary datasets have been generated and deposited for this study.

The source data of this paper are collected in the following database record: biostudies:S-SCDT-10_1038-S44319-025-00547-w.

## Peer review information

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

## Acknowledgements

We thank Yann Denis from the transcriptomic platform for its expertise and technical assistance in quantitative RT-PCR. We thank R. Daniel for the gift of the AG157 strain. This research was supported by the CNRS, the Agence Nationale de la Recherche to AG (ANR-19-CE15-0011-01) and Aix-Marseille University and PPIA 4D-OMICS [21-ESRE-0052] project for providing infrastructure support. This project was also supported by the European Research Council (ERC) under the Horizon 2020 research and innovation program, ERC consolidator grant to RC-L (agreement ID: 772178).

## Author contributions

**Frédérique Pompeo**: Conceptualization; Data curation; Supervision; Investigation; Methodology; Writing—original draft; Writing—review and editing. **Elodie Foulquier**: Investigation. **Arnaud Chastanet**: Data curation; Investigation; Methodology; Writing—original draft. **Leon Espinosa**: Methodology. **Cyrille Billaudeau**: Investigation; Methodology. **Anthony Rodrigues**: Investigation. **Charlène Cornilleau**: Investigation. **Rut Carballido-López**: Funding acquisition; Writing—original draft. **Anne Galinier**: Conceptualization; Funding acquisition; Writing—original draft.

Source data underlying figure panels in this paper may have individual authorship assigned. Where available, figure panel/source data authorship is listed in the following database record: biostudies:S-SCDT-10_1038-S44319-025-00547-w.

## Disclosure and competing interests statement

The authors declare no competing interests.

