## [Peer Review File · EMBO Reports]

RagB stimulates the activity of the peptidoglycan polymerase RodA in *Bacillus subtilis*.

Frédérique Pompeo, Elodie Foulquier, Arnaud Chastanet, Leon Espinosa, Anthony Rodrigues, Charlène Cornilleau, RUT CARBALLIDO LOPEZ, Anne Galinier, and Cyrille Billaudeau

Corresponding author(s): *Frédérique Pompeo* (fpompeo@imm.cnrs.fr)

Review Timeline:

Submission Date:	17th Jan 25
Editorial Decision:	18th Feb 25
Revision Received:	20th Jun 25
Editorial Decision:	16th Jul 25
Revision Received:	23rd Jul 25
Accepted:	29th Jul 25

Editor: Achim Breiling

Transaction Report:

Dear Dr. Pompeo,

Thank you for the submission of your manuscript to EMBO reports. I have now received the reports from the three referees that were asked to evaluate your study, which can be found at the end of this email.

As you will see, the referees think that these findings are of interest. However, they have several comments, concerns, and suggestions, indicating that a major revision of the manuscript is necessary to allow publication of the study in EMBO reports. As the reports are below, and all the referee concerns need to be addressed, I will not detail them here.

Given the constructive referee comments, I would like to invite you to revise your manuscript with the understanding that the concerns of the referees must be addressed in the revised manuscript and in a detailed point-by-point response. Acceptance of your manuscript will depend on a positive outcome of a second round of review. It is EMBO reports policy to allow a single round of revision only and acceptance of the manuscript will therefore depend on the completeness of your responses included in the next, final version of the manuscript.

- 1) a .docx formatted version of the final manuscript text (including legends for main figures, EV figures and tables), but without the figures included. Figure legends should be compiled at the end of the manuscript text.
- 2) individual production quality figure files as .eps, .tif, .jpg (one file per figure), of main figures and EV figures. Please upload these as separate, individual files upon re-submission.

- 4) a complete author checklist, which you can download from our author guidelines (<https://www.embopress.org/page/journal/14693178/authorguide>). Please insert page numbers in the checklist to indicate where the requested information can be found in the manuscript. The completed author checklist will also be part of the RPF.

- 5) that primary datasets produced in this study (e.g. RNA-seq, ChIP-seq, structural and array data) are deposited in an

appropriate public database. If no primary datasets have been deposited, please also state this in a dedicated section (e.g. 'No primary datasets have been generated and deposited'), see below.

The accession numbers and database should be listed in a formal "Data Availability" section that follows the model below. This is now mandatory (like the COI statement). Please note that the Data Availability Section is restricted to new primary data that are part of this study. This section is mandatory. As indicated above, if no primary datasets have been deposited, please state this in this section

Data availability

8) Regarding data quantification and statistics, please make sure that the number "n" for how many independent experiments were performed, their nature (biological versus technical replicates), the bars and error bars (e.g. SEM, SD) and the test used to calculate p-values is indicated in the respective figure legends (also for EV and Appendix figures). Please also check that all the p-values are explained in the legend, and that these fit to those shown in the figure. Please provide statistical testing where applicable. Please avoid the phrase 'independent experiment', but clearly state if these were biological or technical replicates. Please also indicate (e.g. with n.s.) if testing was performed, but the differences are not significant. In case n=2, please show the data as separate datapoints without error bars and statistics. See also: <http://www.embopress.org/page/journal/14693178/authorguide#statisticalanalysis>

9) Please add scale bars of similar style and thickness to microscopic images, using clearly visible black or white bars (depending on the background). Please place these in the lower right corner of the images themselves. Please do not write on or near the bars in the image but define the size in the respective figure legend.

10) Please also note our reference format:

12) We now use CRediT to specify the contributions of each author in the journal submission system. CRediT replaces the author contribution section. Please use the free text box to provide more detailed descriptions and do NOT provide your final manuscript text file with an author contributions section. See also our guide to authors: <https://www.embopress.org/page/journal/14693178/authorguide#authorshipguidelines>

13) All Materials and Methods need to be described in the main text using our 'Structured Methods' format, which is required for

all research articles. According to this format, the Methods section should include a Reagents and Tools Table (listing key reagents, experimental models, software, and relevant equipment and including their sources and relevant identifiers), uploaded as separate file, and a Methods section in which we encourage the authors to describe their methods using a step-by-step protocol format with bullet points, to facilitate the adoption of the methodologies across labs. More information on how to adhere to this format as well as downloadable templates (.doc) for the Reagents and Tools Table can be found in our author guidelines (section 'Structured Methods'):

14) Please order the sections like this, using these names:

Title page - Abstract - Keywords - Introduction - Results - Discussion - Methods - Data availability section - Acknowledgements (including the funding information) - Disclosure and Competing Interests Statement - References - Figure legends - Expanded View Figure legends

Please remove the list of abbreviations, but define each abbreviation the first time it is used in the manuscript main text.

15) Please make sure that all the funding information is also entered into the online submission system and that it is complete and similar to the one in the acknowledgement section of the manuscript text file.

16) Please add a paragraph titled 'Biosafety' to the methods section gathering all information on where and how biosafety-relevant experiments with microbes were performed and that these were approved, and by whom (institution, government).

I look forward to seeing a revised form of your manuscript when it is ready.

Yours sincerely,

Referee #1:

In this paper the authors investigate two genes implicated in PG synthesis. Although deletion of either gene has no observable effect on cell physiology, they observe that deletion of one of these, *yrpS*-renamed *ragB*, affects growth and morphology in the absence of PBP1, somewhat similar to deletion of *GpsB*. They therefore focus on *ragB*. Overall, the work is carefully done and it is clear that there is some connection between *RagB* and *RodA*. However, it is not clear how *RagB* works.

It is shown that deletion of *gpsB* and *ragB* are distinct. Also, it is observed that overexpression of *RodA* suppresses the lack of *ragB* and PBP1, although not as well as overexpression of *RagB* itself. They also show that overexpression of *rodA* or *ragB* suppresses the sensitivity to moenomycin. This is consistent with *rodA* providing GT activity in the absence of PBPA activity and that *RodA* has a basal activity that can be enhanced by *ragB*.

Comments:

RagB does not move with the characteristics of *RodA* which complicates models, for example that *RagB* directly interacts with *RodA*.

The title seems overstated since in the text (e.g. line 341) of the results the authors say *RodA* activity may be dependent on *RagB*. Also, first line of the discussion (line 349) 'suggesting'.

What is the reference for the statement on lines 168-9?

Line 173, change 'than' to 'as'

Do you ever observe suppressors (fast growers) arising in the strain deleted for *ponA* and *ragB*?

Fig. 2. I am surprised that the *rodA* catalytic mutant does not display a dominant negative phenotype when overexpressed. It even appears to help a bit in the growth curves in Fig. 2C. However, in panel D it appears to slow growth some as the plate is less turbid looking.

Line 177. 'restored' should be replaced with 'rescued'

Fig. 3. RodA expression is measured by rtPCR. Are the authors sure that *rodA* is not subject to posttranscriptional control.

Line 253. Replace 'part of' with 'some'. Also, on line 254

Fig. 6. When 6His-RagB was used as bait was the elution checked for anything other than RodA-GFP? For example one the cognate PBPs or MreC?

Line 366. 'Rab may be a component of the elongasome' this is not consistent with the tracking data

Line 403. Drop the 's' from ameliorates

Line 404. Replace 'of' with 'with'

Line 4114. 'close' should be 'closed'

Referee #2:

The paper "RagB stimulates the activity of the peptidoglycan polymerase RodA in *Bacillus subtilis*" is an elegant paper.

This work both discovers and thoroughly demonstrates that RagB acts as an activator of RodA - specifically in RodA's function, where it fills in as an aPBP when cells lack aPBP activity. Overall, this work does a thorough job demonstrating RagB activates RodA activity, and I have no concerns regarding their experiments.

I have only a few minor corrections, but I would also like to bring a more general aspect of these findings to the author's attention, as I believe their experiments.

1. in light of past work, may point to a more conclusive understanding of RagB's role with RodA and how it functions. A lot of the discussion in the paper focuses on RagB's role in activating/altering RodA's activity within the full MreB-associated Rod Complex, which their data indicates is not the case. However, RagB serves to help RodA functionally act as an aPBP (which all of their data points to). Should the primary assumption not be that, just as with aPBPs, RagB/RodA should indeed function outside of the MreB-associated rod complex? Combined with the past work, their findings in this paper firmly point toward that model. aPBPs not only prevent cells from dying, they also serve to widen cells, and correspondingly do not show directional motion, but rather are transiently immobile for ~3 seconds, fitting with the idea that they are inserting non-oriented material in the wall. Correspondingly, all of their data points to the same. 1) RagB/RodA rescues growth, 2) makes cells wider, and 3) RagB has no effect on the MreB-associated rod complex. Thus, it is most likely that RagB/RodA is also not inserting oriented material and thus is independent of MreB, operating just as aPBPs. I mention this only as these associations may allow the authors to make a stronger conclusion from their data as to how RagB acts with RodA to conduct aPBP-like functions.

2. In this light, can the authors examine their TIRF data, looking not for directional motion but rather immobile particles? This would make far more sense given that the RagB/RodZ gives an aPBP-like function and aPBP phenotypes. Seeing such immobile particles would also bolster their conclusion. (I cannot access the movies to see this myself).

3. In regards to the growth rescue. Dion showed that loss of aPBP activity did not affect single-cell growth rate, but rather the death rate of cells, which overall leads to changes in the growth curves. Have the authors examined single-cell growth rates? If the growth rates are similar, this would also give strong evidence for a protective "fill-in-holes" function for RagB/RodZ.

4. I conducted an alpha-fold multimer prediction of RodA and RagB. Interestingly, this predicted the globular domain of RagB sitting on top of RodA. This is available at: <https://www.dropbox.com/scl/fi/cnvcvz8eziogv4p8u3ua/ragB-RodA.pdb?rlkey=f5yeg83zresb7p4hufb3g76ub&dl=0> if the authors are interested.

Small points:

- 1 Can the standard nomenclature of Δ rather than D be used to indicate the knockouts in figure 2?
2. statistical comparisons are needed for Figure 3A
3. statistical comparisons are needed for Figure 5D
4. 5E - the star on the significance between WT and 1694 is a box, this should be corrected to a *.

Referee #3:

In this study, Pompeo et al. describe the RagB (formerly YrrS) protein as an allosteric regulator of RodA, the essential glycosyltransferase (GT) in the elongasome complex for peptidoglycan (PG) synthesis. Overall, the work is thorough, well-documented, and clearly described. This study significantly extends our understanding of RagB function and explores several possible mechanisms that could explain this protein's role.

Major comments:

1. This study does an excellent job of documenting the genetic (Fig. 1,7), physiological (Figs. 1-3), and biochemical (Fig. 6) interactions between RagB and the essential RodA GT enzyme. Collectively, these data document that RagB increases the apparent activity of RodA but does not seem to affect the mRNA or protein level. These aspects are all very clear.

The next logical question is, does RagB increase RodA activity as part of the elongasome? Or when it is freely diffusing (and potentially acting in PG "repair")? Or both? This is addressed in the microscopy studies in Fig. 4 and 5. These are an important addition to the paper, but the interpretation is not completely clear. RagB protein does not seem to have obvious trajectories like those seen for elongasome proteins (l. 247). Moreover, cells with and without RagB have very similar elongasome behavior (l. 280: "RagB has a limited impact on the localization and activity of RodA associated to the Rod complex.").

I suggest that the authors consider a slight reorganization of the paper and perhaps move Figs. 6 and 7 before 4 and 5... establish the strong case that RagB affects RodA activity first (summarized in para. 1 above), and then explore the more detailed question highlighted in para. 2 at the end of the Results...

2. Next, I request that the authors re-visit their Discussion section (and their presentation of Fig. 8) to improve flow. Please use this section to clearly lay out the evidence for and against a role for RagB at the elongasome. The model that RagB functions in the elongasome is presented in the para. starting l. 362, then the alternative model (and the role of the IDR) is in para. starting l. 388, but then the para. starting l. 403 returns to the idea of the elongasome (perhaps integrate this material with the earlier para.?).

3. Please also work to address confusing or contradictory statements:

- l. 365 "We do not know how RagB boosts the elongasome activity". But does it? Or does it act on the diffusive fraction?
- l. 429 "this overproduction requires also the presence of RagB which becomes essential in the absence of aPBPs" This statement seems incorrect. Perhaps reword to indicate that the increase in RodA activity under conditions that inhibit aPBPs requires both upregulation (SigM) and allosteric activation (RagB).

Comments:

l. 157. Do *gpsB* and *ragB* function in the same pathway? Adding in the triple mutant (*ponA ragB gpsB*) in Fig. 1 and comparing to the double mutants would reveal whether these are in the same pathway (i.e. not additive).

l. 197 and Fig. 2D. The moenomycin effect (comparing WT vs. *ragB*) are visible, but would benefit from quantitation and statistical analysis. The first two sentences in the Fig. 2D legend belong in the Methods section. Also, what is meant by "few" microliter?

Fig. 2A and 2B. These panels are hard to decipher since there are so many strains and there is no embedded legend. I recommend keeping the single and double mutants in (A) and including all studies with xylose-induced expression in panel (B). Currently, the two panels are overlapping in content.

Fig. 3. Panel A and B. Define the boxes and error bars in the legend. These results show the lack of impact of *ragB* on *rodA* mRNA. Please summarize the evidence that *ragB* does not affect RodA protein level (is this from microscopy?).

Panel 3C: Perhaps comment in the legend that the signal for *ragB* in lane 7 is due to spillover from lane 6 or use a cleaner western blot.

l. 238. "slightly discontinuous". This type of staining can be meaningful, but can also be a staining artifact. I'm not sure how to distinguish.

Fig. 4F. Readers might benefit from a little more guidance in how to interpret these kymographs. What is significant here?

l. 427. Perhaps mention moenomycin etc. here, since this is more relevant to cell physiology than a gene deletion.

Writing / formatting issues:

Introduction. For ease in reading, I suggest breaking the first long paragraph into several shorter ones (possibly at l. 75 and 87

l. 127. "such genetic" might be clearer as "this genetic"

l. 150. This is the first mention of *ragB* in the Results. Perhaps mention that this gene was formerly *yrrS*.

I. 153. Why an initial study? Maybe the idea here is that this paper will focus on the role RagB

Figure 1C. Change "D" to delta symbol.

II. 196, 429, 770: I recommend changing "overproduced" to "upregulated" (as in line 216). For former is often used to describe an artificial condition (e.g. xyl promoter, or when overproducing a protein for purification. The latter is used when describing a natural regulatory process (in this case through SigM).

Fig. 4F and 5D. correct the in figure labeling on the kymographs.

L. 253. Part of RodA molecules is... change to Some of the RodA molecules are...

I. 267/269. Maybe use a +/- symbol rather than writing as a fraction to avoid confusion

I. 287. Add comma after control.

Point-by-point response to the three referees

Referee #1:

In this paper the authors investigate two genes implicated in PG synthesis. Although deletion of either gene has no observable effect on cell physiology, they observe that deletion of one of these, *yrpS*-renamed *ragB*, affects growth and morphology in the absence of PBP1, somewhat similar to deletion of *GpsB*. They therefore focus on *ragB*. Overall, the work is carefully done and it is clear that there is some connection between *RagB* and *RodA*. However, it is not clear how *RagB* works.

It is shown that deletion of *gpsB* and *ragB* are distinct. Also, it is observed that overexpression of *RodA* suppresses the lack of *ragB* and PBP1, although not as well as overexpression of *RagB* itself. They also show that overexpression of *rodA* or *ragB* suppresses the sensitivity to moenomycin. This is consistent with *rodA* providing GT activity in the absence of PBPA activity and that *RodA* has a basal activity that can be enhanced by *ragB*.

Comments:

RagB does not move with the characteristics of *RodA* which complicates models, for example that *RagB* directly interacts with *RodA*.

We agree with the comment of the referee. Following as well the comments of reviewers 2 & 3, we have explored the possibility that *RodA* and *RagB* interact independently of the Rod complex with different movement characteristics. This aspect has been the subject of dedicated experiments that have been added to the present manuscript in Fig. 7 and Appendix Figure S7, described in the method (from line 606) and result (from line 377) sections and discussed (lines 467 - 479).

The title seems overstated since in the text (e.g. line 341) of the results the authors say *RodA* activity may be dependent on *RagB*. Also, first line of the discussion (line 349) 'suggesting'.

We have kept the title but replaced 'suggest' with a statement (line 407).

What is the reference for the statement on lines 168-9?

We have added (line 181) the reference

"Meeske, A. J. *et al.* SEDS proteins are a widespread family of bacterial cell wall polymerases. *Nature* **537**, 634-638 (2016). <https://doi.org/10.1038/nature19331>".

Line 173, change 'than' to 'as'

Done (line 186).

Do you ever observe suppressors (fast growers) arising in the strain deleted for *ponA* and *ragB*?

We have never observed larger colonies when we spread the strain $\Delta\textit{ponA}\Delta\textit{gpsB}$ on plates. It grows very slowly and the colonies are very small. Perhaps the dish should be left longer at 37°C to observe any suppressors.

Fig. 2. I am surprised that the *rodA* catalytic mutant does not display a dominant negative phenotype when overexpressed. It even appears to help a bit in the growth curves in Fig. 2C. However, in panel D it appears to slow growth some as the plate is less turbid looking.

As RodA is essential, the wild-type copy of the gene was retained in the overproduction strains. This could potentially explain why the RodA catalytic mutant does not display a dominant negative phenotype when overexpressed. A certain amount of wild-type protein is present, although probably less than that of the overproduced mutated protein.

Line 177. 'restored' should be replaced with 'rescued'
Done (line 190).

Fig. 3. RodA expression is measured by rtPCR. Are the authors sure that *rodA* is not subject to posttranscriptional control.

As we do not have any anti-RodA antibodies, we were unable to estimate the amount of RodA in the different strains in order to check a possible post-transcriptional regulation. However, we did a western-blot with anti-Halo-tag antibodies on crude extracts of strains used in microscopy and producing Halo-RodA from the natural promoter. These results are presented in supplementary data (Appendix Fig. S4). The specificity of the antibodies seems good since no signal is detected in a crude extract of strain without Halo-tag protein (lane 1). However, several bands can be observed at a size compatible with Halo-RodA, or partially degraded Halo-RodA, bearing in mind that membrane proteins do not always migrate at the expected size. Nevertheless, as expected, there seems to be more RodA in the $\Delta\textit{ponA}$ and a little more in the $\Delta\textit{ponA} \Delta\textit{ragB}$ mutants (may be due to increased damages to the cell wall compared to $\Delta\textit{ponA}$) compared to the WT strain but unchanged in $\Delta\textit{ragB}$ mutant. These observations are consistent with the RT-PCR data and suggest that there is no post-transcriptional control of RodA by RagB. Given the poor quality of the western blot, we preferred to add this data as a supplemental figure and not add it to Fig. 3.

Line 253. Replace 'part of' with 'some'. Also, on line 254
Done (lines 346-347).

Fig. 6. When 6His-RagB was used as bait was the elution checked for anything other than RodA-GFP? For example one the cognate PBPs or MreC?

Here, the specific co-elution of GFP-RodA was checked by western-blot with anti-GFP antibodies. However, when we analyzed the co-elution fraction by mass spectrometry, we found known RagB partners such as GpsB and MreC or proteins whose function is linked to PG synthesis like PBP4, DacA,

DapX or FtsE. However, the mass spectrometry analysis was only reproduced twice and we felt that this information was not strictly necessary for the message of the paper and it has not been added.

Line 366. 'RagB may be a component of the elongasome' this is not consistent with the tracking data. If we look at the microscopy data, it's true that the hypothesis 'RagB may be a component of the elongasome' is unlikely. However, as RagB seems to interact with several elongasome proteins (GpsB, RodZ, MreC, PBP2A), it didn't seem right to rule out this possibility entirely. We have tried to give the arguments supporting each of the 2 hypotheses in the discussion.

Line 403. Drop the 's' from ameliorates

Done (line 445).

Line 404. Replace 'of' with 'with'

Done (line 446).

Line 414. 'close' should be 'closed'

Done (line 456).

Referee #2:

The paper "RagB stimulates the activity of the peptidoglycan polymerase RodA in *Bacillus subtilis*" is an elegant paper.

This work both discovers and thoroughly demonstrates that RagB acts as an activator of RodA - specifically in RodA's function, where it fills in as an aPBP when cells lack aPBP activity. Overall, this work does a thorough job demonstrating RagB activates RodA activity, and I have no concerns regarding their experiments.

I have only a few minor corrections, but I would also like to bring a more general aspect of these findings to the author's attention, as I believe their experiments.

1. in light of past work, may point to a more conclusive understanding of RagB's role with RodA and how it functions. A lot of the discussion in the paper focuses on RagB's role in activating/altering RodA's activity within the full MreB-associated Rod Complex, which their data indicates is not the case. However, RagB serves to help RodA functionally act as an aPBP (which all of their data points to). Should the primary assumption not be that, just as with aPBPs, RagB/RodA should indeed function outside of the MreB-associated rod complex? Combined with the past work, their findings in this paper firmly point toward that model. aPBPs not only prevent cells from dying, they also serve to widen cells, and correspondingly do not show directional motion, but rather are transiently immobile for ~3 seconds, fitting with the idea that they are inserting non-oriented material in the wall.

Correspondingly, all of their data points to the same. 1) RagB/RodA rescues growth, 2) makes cells wider, and 3) RagB has no effect on the MreB-associated rod complex. Thus, it is most likely that RagB/RodA is also not inserting oriented material and thus is independent of MreB, operating just as aPBPs. I mention this only as these associations may allow the authors to make a stronger conclusion from their data as to how RagB acts with RodA to conduct aPBP-like functions.

We thank the reviewer for these very interesting thoughts, that was shared as well by the third reviewer (his point 3). The possibility that RagB/RodA could work outside the Rod complex, inserting material in a non-directional manner similarly to aPBPs, is a good suggestion that could not be answered based on our previous data. To address this, as explained below in our response to point 2, we have conducted new experiments that have been added to the current manuscript, and we have accordingly modified the result and method sections and updated our discussion.

2. In this light, can the authors examine their TIRF data, looking not for directional motion but rather immobile particles? This would make far more sense given that the RagB/RodA gives an aPBP-like function and aPBP phenotypes. Seeing such immobile particles would also bolster their conclusion. (I cannot access the movies to see this myself).

We are deeply sorry that the movies were not accessible (unclear why since they were loaded on the server accordingly to the editor recommendations) and hope they will be this time. They would not have answered the question though because the density of RodA foci were so high that they prevented the tracking and classification of patch dynamics as we usually do. This is why we originally had to turn to kymograph analyses rather than MSD, using long exposure times, to reveal the presence of the directional foci.

To overcome this issue and quantify the dynamic of individual foci to address this hypothesis, we turned to single particle tracking using limited amount of Halo ligand in order to turn on only a fraction of the RodA-Halo population. This allowed us to performed both MSD and CDF analysis. The latter was used in previous studies (Cho et al., 2016, Vigouroux et al., 2020) that lead to the hypothesis that the slow-moving (or static in Vigouroux et al., 2020) subpopulation of aPBPs is the fraction actively inserting (non-directional) PG. This method was thus applied here to detect the existence of a similar static or slow-moving subpopulation of RodA, that would be affected by the absence of RagB.

The results are now presented as Fig. 7 panel G & H and Appendix Fig. S7. To summarize our results, we were indeed able to identify two subpopulations of diffusing RodA particles with a 20/80 ratio between slow and fast foci. The absence of *ponA* slightly increased the proportion of slow moving RodA, which advocates for the hypothesis that RodA might concur to non-directional PG synthesis, similarly to aPBPs. The absence of RagB alone had no impact on the ratio or on the diffusion coefficient of this subpopulation, but a limited impact on the fraction of slow moving RodA was observed in absence of both *ragB* and *ponA*.

Overall, the effects observed, previously on the directional RodA and now on the slow diffusing RodA (in both cases in the absence of *ponA*), are however of limited magnitudes. We have nonetheless updated our discussion to integrate these new results and the hypothesis on the role of RagB and RodA acting outside of the Rod complex.

3. In regards to the growth rescue. Dion showed that loss of aPBP activity did not affect single-cell growth rate, but rather the death rate of cells, which overall leads to changes in the growth curves.

Have the authors examined single-cell growth rates? If the growth rates are similar, this would also give strong evidence for a protective "fill-in-holes" function for RagB/RodA.

As suggested, we did liquid cultures to follow bulk growth using a microplate reader and time-lapses on agar PAD using microscopy to follow single-cell division rates as done in the paper from "Dion et al. 2019" in Fig. 2d, excepted that we did growth in LB medium with 0.5 % xylose. We compared growth rates for the WT strain, $\Delta 4$, and $\Delta 4$ or $\Delta 4\Delta ragB$ mutants complemented by RodA overproduction (strains WT, AG157, SG1150 and SG1158 respectively). The results presented below revealed that we were not able to reproduce the results obtained in the publication for the strains WT, $\Delta 4$ and $\Delta 4$ +RodA in which they showed that the increased rate of cell death in the $\Delta 4$ aPBPs was suppressed by RodA overproduction, leading to higher bulk growth while the single cell growth rate remained constant. Indeed, in our conditions (growth in LB instead of CH medium), we observed that growth of the $\Delta 4$ strain is strongly affected at both population and single-cell levels and is not complemented by the overproduction of RodA, contrary to what we observed on solid media at the level of the bacterial colony (Figure 5A). In the time-lapse experiments for $\Delta 4$ and $\Delta 4$ +RodA strains (AG157 and SG1150), we could clearly see that some bacteria were not dividing at all and some others were dividing much more slowly than the WT cells. However, in the absence of RagB (strain SG1158), growth rates of the cell population and the single-cell are both much more affected (see below) as previously observed on solid media, even in the presence of magnesium (Figure 5A). It is also important to remember that the strain $\Delta 4\Delta ragB$ is not viable unless RodA is overproduced.

We can therefore confirm that RagB is important for the activity of RodA but without being able to specify whether it is regulated only during the activity required for PG repair ("fill-in-holes" function) or that required for cell elongation, or both.

Strains	μ (average) H^{-1}
WT	1,54
AG157 ($\Delta 4$)	0,54
SG1150 ($\Delta 4 + \text{RodA}$)	0,51
SG1158 ($\Delta 4 + \Delta \text{ragB} + \text{RodA}$)	0,12

Strains	μ (average) H^{-1}
WT	2,03
AG157 ($\Delta 4$)	1,43
SG1150 ($\Delta 4 + \text{RodA}$)	1,27
SG1158 ($\Delta 4 + \Delta \text{ragB} + \text{RodA}$)	0,85

4. I conducted an alpha-fold multimer prediction of RodA and RagB. Interestingly, this predicted the globular domain of RagB sitting on top of RodA. This is available at:

<https://www.dropbox.com/scl/fi/cnvcvz8eziovg4p8u3ua/ragB-RodA.pdb?rlkey=f5yeg83zresb7p4hufb3g76ub&dl=0> if the authors are interested.

We thank the referee for this model. We had also tested it ourselves but the flexibility of the IDR domain of RagB makes it unreliable. It therefore seems difficult to use it to support hypotheses although this model is encouraging if we imagine that RagB targets RodA to the sites of PG repair via its IDR.

Small points:

1. Can the standard nomenclature of Δ rather than D be used to indicate the knockouts in figure 2?

Done.

2. statistical comparisons are needed for Figure 3A

Done.

3. statistical comparisons are needed for Figure 5D

We think the referee means Fig. 5E. The differences are non-significant. This has been added to the figure.

4. 5E - the star on the significance between WT and 1694 is a box, this should be corrected to a *.

Done.

Referee #3:

In this study, Pompeo et al. describe the RagB (formerly YrrS) protein as an allosteric regulator of RodA, the essential glycosyltransferase (GT) in the elongasome complex for peptidoglycan (PG) synthesis. Overall, the work is thorough, well-documented, and clearly described. This study significantly extends our understanding of RagB function and explores several possible mechanisms that could explain this protein's role.

Major comments:

1. This study does an excellent job of documenting the genetic (Fig. 1,7), physiological (Figs. 1-3), and biochemical (Fig. 6) interactions between RagB and the essential RodA GT enzyme. Collectively, these data document that RagB increases the apparent activity of RodA but does not seem to affect the mRNA or protein level. These aspects are all very clear. The next logical question is, does RagB increase RodA activity as part of the elongasome? Or when it is freely diffusing (and potentially acting in PG "repair")? Or both? This is addressed in the microscopy studies in Fig. 4 and 5. These are an important addition to the paper, but the interpretation is not completely clear. RagB protein does not seem to have obvious trajectories like those seen for elongasome proteins (l. 247). Moreover, cells with and without RagB have very similar elongasome behavior (l. 280: "RagB has a limited impact on the localization and activity of RodA associated to the Rod complex.").

I suggest that the authors consider a slight reorganization of the paper and perhaps move Figs. 6 and 7 before 4 and 5... establish the strong case that RagB affects RodA activity first (summarized in para. 1 above), and then explore the more detailed question highlighted in para. 2 at the end of the Results...

We have followed the referee's advice and inverted the sections of the manuscript so that the different approaches showing the role of RagB on RodA activity are grouped together at the beginning of the results, followed by the microscopy data.

2. Next, I request that the authors re-visit their Discussion section (and their presentation of Fig. 8) to improve flow. Please use this section to clearly lay out the evidence for and against a role for RagB at the elongasome. The model that RagB functions in the elongasome is presented in the para. starting l. 362, then the alternative model (and the role of the IDR) is in para. starting l. 388, but then the para. starting l. 403 returns to the idea of the elongasome (perhaps integrate this material with the earlier para.?).

We modified the organization of the several sections of the discussion in order to make it smoother and easier to follow and tried to display the evidence for and against a role for RagB at the elongasome.

3. Please also work to address confusing or contradictory statements:
l. 365 "We do not know how RagB boosts the elongasome activity". But does it? Or does it act on the diffusive fraction?

We replaced "elongasome activity" by "RodA activity" in this sentence in order to avoid contradictory statements. The potential role of RagB in and outside of the rod complex has been the subject of new experiments, and generating new results that have now been added and discussed in the present manuscript, as explained above on our answer to comments point 2 of Reviewer #2.

l. 429 "this overproduction requires also the presence of RagB which becomes essential in the absence of aPBPs" This statement seems incorrect. Perhaps reword to indicate that the increase in RodA activity under conditions that inhibit aPBPs requires both upregulation (SigM) and allosteric activation (RagB).

We changed this sentence as suggested to avoid incorrect statement (line 506).

Comments:

l. 157. Do *gpsB* and *ragB* function in the same pathway? Adding in the triple mutant (*ponA ragB gpsB*) in Fig. 1 and comparing to the double mutants would reveal whether these are in the same pathway (i.e. not additive).

We have added in the Appendix Fig. S3A a comparison of the growth of the triple mutant with that of the double mutants. This shows an additive effect of the deletions, suggesting that the two proteins RagB and GpsB are involved in different pathways. However, these results do not rule out their possible role in a same function. We have added this result to the manuscript (line 204).

In addition, we looked at complementation of the triple mutant ($\Delta ponA \Delta ragB \Delta gpsB$) by overproduction of PBP1, RagB, GpsB or RodA (Appendix Fig. S3B and C). Only the overproduction of PBP1 allowed a return to normal growth, which is logical since the double mutant ($\Delta ragB \Delta gpsB$) has no phenotype. The lack of complementation by overproduction of RagB or GpsB confirms that the two proteins are also involved in different pathways. The lack of complementation by overproduction of RodA was therefore expected.

l. 197 and Fig. 2D. The moenomycin effect (comparing WT vs. *ragB*) are visible, but would benefit from quantitation and statistical analysis. The first two sentences in the Fig. 2D legend belong in the Methods section. Also, what is meant by "few" microliter?

Quantitation of the moenomycin effect and statistical analysis were made and added to Fig. 2.

We used 'few' because the volume of pre-culture used for seeding the main culture varied slightly according to the OD of the strain. This ensures the same initial OD for all strains cultures. But this has been changed in the Methods section.

Fig. 2A and 2B. These panels are hard to decipher since there are so many strains and there is no embedded legend. I recommend keeping the single and double mutants in (A) and including all studies with xylose-induced expression in panel (B). Currently, the two panels are overlapping in content.

Done.

Fig. 3. Panel A and B. Define the boxes and error bars in the legend. These results show the

lack of impact of ragB on rodA mRNA. Please summarize the evidence that ragB does not affect RodA protein level (is this from microscopy?).

Boxes and error bars (named SD) have been defined in the legend.

As we do not have any anti-RodA antibodies, we were unable to estimate the amount of RodA in the different strains in order to check a possible post-transcriptional regulation. However, we did a western-blot with anti-Halo-tag antibodies on crude extracts of strains used in microscopy and producing Halo-RodA from the natural promoter. These results are presented in supplementary data (Appendix Fig. S4). The specificity of the antibodies seems good since no signal is detected in a crude extract of strain without Halo-tag protein (lane 1). However, several bands can be observed at a size compatible with Halo-RodA, or partially degraded Halo-RodA, bearing in mind that membrane proteins do not always migrate at the expected size. Nevertheless, as expected, there seems to be more RodA in the $\Delta ponA$ and a little more in the $\Delta ponA \Delta ragB$ mutants (may be due to increased damages to the cell wall compared to $\Delta ponA$) compared to the WT but unchanged in $\Delta ragB$ mutant. These observations are consistent with the RT-PCR data and suggest that there is no post-transcriptional control of RodA by RagB. Given the poor quality of the western blot, we preferred to add this data as a supplemental figure and not add it to Fig. 3.

Panel 3C: Perhaps comment in the legend that the signal for ragB in lane 7 is due to spillover from lane 6 or use a cleaner western blot.

A cleaner western blot has been used.

I. 238. "slightly discontinuous". This type of staining can be meaningful, but can also be a staining artifact. I'm not sure how to distinguish.

We removed "slightly discontinuous" to avoid misunderstanding (line 330). In Fig. 4, RagB is not stained with a fluorescent molecule, we used a GFP-RagB fusion. We observed non-homogeneous distribution of the protein in the membrane.

Fig. 4F. Readers might benefit from a little more guidance in how to interpret these kymographs. What is significant here?

We added a sentence in the legend of the figure explaining that non obvious trajectories were observed.

I. 427. Perhaps mention moenomycin etc. here, since this is more relevant to cell physiology than a gene deletion.

Moenomycin effect has been mentioned (line 502).

Writing / formatting issues:

Introduction. For ease in reading, I suggest breaking the first long paragraph into several shorter ones (possibly at l. 75 and 87)

Done (lines 76 and 92).

I. 127. "such genetic" might be clearer as "this genetic"

Done (line 137).

I. 150. This is the first mention of ragB in the Results. Perhaps mention that this gene was formerly yrrS.

Done (line 160).

I. 153. Why an initial study? Maybe the idea here is that this paper will focus on the role RagB

Yes, we removed "initial" (line 165).

Figure 1C. Change "D" to delta symbol.

Done.

II. 196, 429, 770: I recommend changing "overproduced" to "upregulated" (as in line 216). For former is often used to describe an artificial condition (e.g. xyl promoter, or when overproducing a protein for purification. The latter is used when describing a natural regulatory process (in this case through SigM).

Done (lines 216, 501 and 1051).

Fig. 4F and 5D. correct the in figure labeling on the kymographs.

Done.

L. 253. Part of RodA molecules is... change to Some of the RodA molecules are...

Done (lines 346 - 347).

I. 267/269. Maybe use a +/- symbol rather than writing as a fraction to avoid confusion

Done.

I. 287. Add comma after control.

Done.

Dear Dr. Pompeo,

Thank you for the submission of your revised manuscript to our editorial offices. I have now received the reports from the three referees that I asked to re-evaluate the study, you will find below. As you will see, the referees now support the publication of your study in EMBO reports. Referees #1 and #3 have some comments and suggestions to improve the manuscript, I ask you to address in a final revised manuscript. Please also provide a final p-b-p-response to these points and my editorial requests below.

Editorial requests:

- Please provide the abstract written in present tense throughout and with no more than 175 words.
- Please order the manuscript sections like this, using these names:
Title page - Abstract - Keywords - Introduction - Results - Discussion - Methods - Data availability section - Acknowledgements - Disclosure and Competing Interests Statement - References - Figure legends
- The data availability section (DAS) is restricted to externally deposited datasets generated in a study. If no primary datasets have been deposited, please state this in this section ("No large primary datasets have been generated and deposited for this study"). Please remove all other information from this section.
- Please add the primer information mentioned in the DAS to the reagents and tools table. All primers used in this study (i.e. their sequences) need to be listed in the reagents and tools table. Please also add call-outs to the table where appropriate.
- Please remove the mention of the Appendix after the DAS.
- Please add the information provided in Table 1 to the reagents and tools table and remove Table 1 from the manuscript. Please update related call-outs.
- Please provide a complete author checklist with responses for cells 106-108.
- Please add scale bars of similar style and thickness to microscopic images, using clearly visible black or white bars (depending on the background). Please place these in the lower right corner of the images themselves. Please do not write on or near the bars in the image but define the size in the respective figure legend. Presently, scale bars have text nearby. Please check.
- Please check again that the number "n" for how many independent experiments were performed, their nature (biological versus technical replicates), the bars and error bars (e.g. SEM, SD) and the test used to calculate p-values is indicated in the respective figure legends. Please also check that all the p-values are explained in the legend, and that these fit to those shown in the figure. Please provide statistical testing where applicable. Please avoid the phrase 'independent experiment' but clearly state if these were biological or technical replicates. Please also indicate (e.g. with n.s.) if testing was performed, but the differences are not significant. In case n=2, please show the data as separate datapoints without error bars and statistics. See also:
<http://www.embopress.org/page/journal/14693178/authorguide#statisticalanalysis>
- If n<5, please show single datapoints for diagrams. Moreover:
 - Please define the annotated p values ****/***/**/* as well as provide the exact p-values for the same in the legend of figure 7F as appropriate.
 - Please note that the exact p values are not provided in the legends of figures 1C, 2D, 3A, 5C
 - Please indicate the statistical test used for data analysis in the legend of figure 5C
 - Please note that the box plots need to be defined in terms of minima, maxima, bounds of box and whiskers, and percentile in the legends of figures 1C, 5C.
 - Please note that the box plots need to be defined in terms of minima, maxima, centre, bounds of box and whiskers, and percentile in the legends of figures 3A, B.
 - Please note that the error bars are not defined in the legend of figure 2D.
- Please add to each legend (main, and EV figures, where applicable) a 'Data Information' section (or name the provided section like this) explaining the statistics used or providing information regarding replicates and scales. See:

- Please make sure that all the funding information is also entered into the online submission system and that it is complete and similar to the one in the acknowledgement section of the manuscript text file. The information presently provided in the Comments box in the submission system (CNRS, Aix-Marseille University, PPIA 4D-OMICS [21-ESRE-0052] project) needs to

be removed from the box and entered as a separate funder (our publisher retrieves the info from the separate entries, not from the box). Please do that.

- Please name the Appendix figures 'Appendix Figure Sx' in the appendix file and make sure they are called out like that.
- Please name the movie files 'Movie EV1', 'Movie EV2' or 'Movie EV3' in all places (source file names, titles in the submission system, their callouts and legends). Please provide each legend as a readme.txt file that then should be ZIPPed up with its corresponding movie and uploaded (so that we have one zip folder per movie). Finally, please remove the movie legends from the Appendix file (and their mention in the table of contents).
- During our figure integrity check, we noted that there is panel reuses between Figure 1B and Figure 2B (delta ponA). In Fig. 2B the image is rotated by 90 degrees. If this reuse is intentional, please clearly indicate this in the respective legends.

In addition, I would need from you uploaded separately:

I look forward to seeing the final revised version of your manuscript when it is ready.

Please let me know if you have questions regarding the revision.

Referee #1:

The authors have responded to the critiques. The paper shows that RagB enhances the activity of RodA although the mechanism is not clear.

Minor comments:

Line 216 "what" do you mean "which"

Line 261 "non-retained" people usually use "flow-thru"

Line 266 what is meant by "denaturation of the interaction with antibodies". This could be made clearer.

Line 504. You might replace "ways" with "modes"

Fig. 6b. What do you make of the RagB localization to the septum? Clearly this must be much different than RodA, consistent with the lack of colocalization.

Referee #2:

All of my concerns have been addressed, and the paper is better than ever. Congratulations on such a great work.

Referee #3:

Overall, the authors have done a very good job of addressing the prior critiques and the result is a stronger and easier to read paper. This study succeeds in identifying a new activator of RodA activity and suggests several possible mechanisms for the observed effects. I have mostly minor suggestions for final revision.

Comments

The authors may be interested to know that *yrpS* (*ragB*) showed up as a gene up-regulated upon induction of σ^M (either a direct or indirect target of regulation) (see PMID: 12644242) This makes sense since σ^M also activates synthesis of RodA and other elongasome components.

I. 92. I am unclear on how aPBPs contribute to recycling?

I. 145. Is the word "necessary" justified here? It is clear that there is a strong defect, but do cells not grow? The double mutant seems to be viable. Alternatively, one could state "RagB supports normal growth and morphology in the absence of PBP1".

I. 179. Is the word "complemented" correct in this context?

I. 408. There may be a more conservative way to phrase this since no direct measurements of GT activity were made.

I. 425. Change freely to free....

Model figure 8. What evidence is there that CW repair by aPBPs is related to MreB class proteins? For the cell wall repair pathway proposed for RodA/bPBP it might be clearer if MreB class proteins were omitted (if they are present, this would seem to correspond to an elongasome). Or, are you saying that not all elongasome com

Point-by-point response to the referees

We would like to thank the referees for their comments and positive feedback.

Referee #1:

The authors have responded to the critiques. The paper shows that RagB enhances the activity of RodA although the mechanism is not clear.

Minor comments:

Line 216 "what" do you mean "which": corrected

Line 261 "non-retained" people usually use "flow-thru": corrected

Line 266 what is meant by "denaturation of the interaction with antibodies". This could be made clearer: this means that the interaction between the GFP protein and the antibody is broken by protein denaturation. The word "denaturation" has been replaced by "disruption".

Line 504. You might replace "ways" with "modes": corrected

Fig. 6b. What do you make of the RagB localization to the septum? Clearly this must be much different than RodA, consistent with the lack of colocalization: Using TIRF, we observed that RagB is not evenly distributed in the membrane but forming constantly reorganizing loosely-defined clusters over the cell periphery. So, on Fig. 6B, it is likely that the double cell wall thickness of the septum concentrates more RagB molecules, leading to a more intense fluorescent signal.

Referee #2:

All of my concerns have been addressed, and the paper is better than ever. Congratulations on such a great work.

Thank you for your congratulations and your previous comments, which have greatly improved the paper.

Referee #3:

Overall, the authors have done a very good job of addressing the prior critiques and the result is a stronger and easier to read paper. This study succeeds in identifying a new activator of RodA activity and suggests several possible mechanisms for the observed effects. I have mostly minor suggestions for final revision.

Thank you for your previous and current suggestions, which have significantly improved the paper.

Comments

The authors may be interested to know that *yrpS* (*ragB*) showed up as a gene up-regulated upon induction of σ^M (either a direct or indirect target of regulation) (see PMID: 12644242) This makes sense since σ^M also activates synthesis of RodA and other elongasome components.

Thanks for the reference. These data are indeed in line with the effects observed here.

I. 92. I am unclear on how aPBPs contribute to recycling?

We meant cell wall maintenance with CW cleavages and CW insertions rather than recycling. The word "recycling" has been changed by "maintenance".

I. 145. Is the word "necessary" justified here? It is clear that there is a strong defect, but do cells not grow? The double mutant seems to be viable. Alternatively, one could state "RagB supports normal growth and morphology in the absence of PBP1".

We removed this part of the sentence "and therefore in which RodA activity becomes necessary for PG synthesis".

I. 179. Is the word "complemented" correct in this context?:

We replaced it by "compensated".

I. 408. There may be a more conservative way to phrase this since no direct measurements of GT activity were made.

We added "most likely" in the sentence.

I. 425. Change freely to free.....: done

Model figure 8. What evidence is there that CW repair by aPBPs is related to MreB class proteins? For the cell wall repair pathway proposed for RodA/bPBP it might be clearer if MreB class proteins were omitted (if they are present, this would seem to correspond to an elongasome). Or, are you saying that not all elongasome com

In the paper from Kawai et al*, they show by several ways, that MreB and PBP1 proteins interact and that PBP1 is delocalized in the absence of MreB. PBP1 localization is independent of the MreB paralogues, Mbl and MreBH but since MreB, MreBH and Mbl colocalize in a single helical structure, we propose to modify the lower left panel of Fig 8, with MreB in bold and MreBH and Mbl clearer with a question mark.

We agree that it might be clearer if MreB class proteins were omitted to the cell wall repair pathway proposed for RodA/bPBP and we removed them from the corresponding part of the diagram (lower right panel).

* Kawai Y, Daniel RA, Errington J. Regulation of cell wall morphogenesis in *Bacillus subtilis* by recruitment of PBP1 to the MreB helix. Mol Microbiol. 2009 Mar;71(5):1131-44. doi: 10.1111/j.1365-2958.2009.06601.x. Epub 2009 Jan 29. PMID: 19192185.

Dr. Frédérique Pompeo
French National Centre for Scientific Research
31 chemin J. Aiguier
Marseille 13009
France

Dear Dr. Pompeo,

Thank you for the submission of your final revised manuscript to our editorial offices. I now went through it and your final p-b-p-response and consider the remaining points of the referees as adequately addressed.

I thus am very pleased to accept your manuscript for publication in the next available issue of EMBO reports. Thank you for your contribution to our journal.

Yours sincerely,
